# Single-cell RNA-seq reveals that glioblastoma recapitulates a normal neurodevelopmental hierarchy

Charles P. Couturier[1], Shamini Ayyadhury [1], Phuong U. Le[1], Javad Nadaf[1,2,3], Jean Monlong [2], Gabriele Riva[1], Redouane Allache[1], Salma Baig[1], Xiaohua Yan[1], Mathieu Bourgey[2,3,4], Changseok Lee[1], Yu Chang David Wang[2,3], V. Wee Yong [5], Marie-Christine Guiot[6], Hamed Najafabadi [2,3], Bratislav Misic[1], Jack Antel[1], Guillaume Bourque [2,3,4], Jiannis Ragoussis [2,3,7] & Kevin Petrecca [1✉]

Cancer stem cells are critical for cancer initiation, development, and treatment resistance. Our understanding of these processes, and how they relate to glioblastoma heterogeneity, is limited. To overcome these limitations, we performed single-cell RNA sequencing on 53586 adult glioblastoma cells and 22637 normal human fetal brain cells, and compared the lineage hierarchy of the developing human brain to the transcriptome of cancer cells. We find a conserved neural tri-lineage cancer hierarchy centered around glial progenitor-like cells. We also find that this progenitor population contains the majority of the cancer's cycling cells, and, using RNA velocity, is often the originator of the other cell types. Finally, we show that this hierarchal map can be used to identify therapeutic targets specific to progenitor cancer stem cells. Our analyses show that normal brain development reconciles glioblastoma development, suggests a possible origin for glioblastoma hierarchy, and helps to identify cancer stem cell-specific targets.

[1] Department of Neurosciences, Montreal Neurological Institute-Hospital, McGill University, Montreal, QC, Canada. [2] Department of Human Genetics, McGill University, Montreal, QC, Canada. [3] McGill University and Genome Québec Innovation Centre, Montreal, QC, Canada. [4] Canadian Centre for Computational Genomics, McGill University, Montreal, QC, Canada. [5] Department of Clinical Neurosciences, University of Calgary, Calgary, AB, Canada. [6] Department of Neuropathology, Montreal Neurological Institute and Hospital, McGill University, Montreal, QC, Canada. [7] Department of Bioengineering, McGill University, Montreal, QC, Canada. ✉email: kevin.petrecca@mcgill.ca

Significant obstacles hampering the development of effective cancer therapeutics include tumor heterogeneity[1–5], and the persistence of incompletely understood cancer stem cells (CSCs) that give rise to cancer recurrence[6,7].

IDH wild-type (IDHwt) glioblastoma, the most common adult primary brain cancer[8], exemplifies these obstacles. Following radiotherapy and temozolomide (TMZ) chemotherapy, the median time to recurrence is 7 months, with patients succumbing to the disease 7 months thereafter[9,10]. This cancer is composed of two main cell compartments: a larger differentiated cell compartment that forms the basis of our understanding of the genomic and molecular underpinnings of the disease[11,12]; and a smaller, less well-characterized compartment of cells with stem-like capabilities[13–16]. The molecular and genomic heterogeneity, and the persistence of a subpopulation of cancer cells with stem-like properties following radiotherapy and chemotherapy, are believed to be the main causes of resistance to treatment and the associated extremely poor outcomes[6,17,18].

Interpatient heterogeneity was established through genomic and transcriptomic analyses by The Cancer Genome Atlas (TCGA) research network[11]. Analysis of whole-tumor transcriptomic data extracted from predominantly differentiated cells showed that glioblastoma clustered into four main subtypes: proneural; neural; classical; and mesenchymal[19]. The more recent classification now excludes the neural subtype[20]. Despite very different transcriptomic profiles and associated genomic alterations, no differences in survival exist between these subtypes. More recently, it has been shown that multiple subtypes coexist in different regions[21,22] and different cells[12,20] within the same tumor. This interpatient and intratumoral heterogeneity poses a daunting challenge for research programs aimed at developing targeted therapeutic approaches[23] and may explain the failures of such approaches in this disease. Although a neurodevelopmental bi-lineage hierarchy has been shown to explain a portion of this heterogeneity in IDH mutant glioma[24,25] and high-grade pediatric glioma[26], this has not been possible in adult IDHwt glioblastoma.

Another layer of complexity was uncovered by the discovery of a small subpopulation of glioblastoma cells that have stem-like properties[13,14]. The CSC theory is derived from our understanding of normal stem cells[15] and posits that such cells must exhibit properties of self-renewal and the ability to produce differentiated progeny. Consistently, glioblastoma stem cells (GSCs) do possess these properties. GSCs can propagate tumors from one host to another[14], and can expand and develop to form brain cancers in orthotopic xenograft models that recapitulate the tumor from which they were extracted[14,27]. Importantly, stem cells isolated from different tumors show variability with respect to marker expression[28–30], suggesting that some degree of interpatient and/or intratumoral heterogeneity exists within the stem cell compartment as well. Although the GSC compartment is small in comparison with the differentiated compartment, it is relevant clinically. Studies have shown that GSCs resist radiotherapy[6] and TMZ chemotherapy[18,31]. These data suggest that GSCs may have a role in cancer development and recurrence. There are presently no treatments targeting GSCs.

Our understanding of glioblastoma heterogeneity, and the relevance of GSCs in this process, is limited. Here, using massively parallel single-cell RNA-sequencing (scRNAseq) of glioblastoma and the normal developing human brain, we discovered a conserved trilineage cancer hierarchy with progenitor cancer cells at the apex. We found that this progenitor population contains the majority of the cancer's cycling cells, corresponds to the apex of the hierarchy using RNA velocity, and functionally resemble GSCs. Clinically relevant, we show that this hierarchal map can be used to identify therapeutic targets specific to GSCs.

## Results

### ScRNAseq highlights genomic heterogeneity in glioblastoma.
We used droplet-based scRNAseq[32–34] to obtain the transcriptome of cells isolated from freshly excised IDHwt glioblastoma and freshly derived enriched GSCs from IDHwt glioblastoma. In total, 53,586 cells from 16 patients (mean age: 62.3 years (95% CI: 57.0, 67.5); 25% female, Table 1) were sequenced: 30,205 whole-tumor cells and 23,381 enriched GSCs.

To distinguish cancer cells from normal brain cells, we determined the main copy number aberration (CNA) events in each cell from its transcriptomic profile (Supplementary Fig. 1b, c). Two clusters devoid of known recurrent CNAs, and containing cells from almost all tumors, were identified (Fig. 1a and Supplementary Fig. 1d). Cells in these clusters expressed genes found exclusively in myeloid cells, oligodendrocytes, or endothelial cells (Supplementary Fig. 1e) and were thus classified as normal cells. All other clusters were formed by cells originating mainly from one to three tumors and contained multiple CNAs. We defined these as cancer cells. When enriched GSCs and whole-tumor cells were sequenced from the same patient, these samples clustered together (Fig. 1a and Supplementary Fig. 1f).

Occasionally, cells from a given patient generated two or three cancer groupings by t-distributed stochastic neighbor embedding (tSNE), likely indicating different clones within a tumor (Fig. 1a). To better characterize these clones, we pooled cells from the cancer clusters of each tumor and reclustered them with our location-averaged data. We determined the correct number of clusters by finding the most-stable solution (Supplementary Fig. 1g). We detected one to three clones for each tumor. These clusters differed by a limited number of CNAs (Supplementary Fig. 1h). Together, these findings demonstrate intertumoural and intratumoral genomic heterogeneity.

### Conserved neurodevelopmental lineages in glioblastoma.
We then assessed intratumoral heterogeneity in the whole-tumor and GSC samples based on single-cell transcriptomic data. We performed principal component analysis (PCA) for GSC samples, and PCA and clustered non-negative matrix factorization (cNMF)[35] for whole-tumor samples to better understand the signatures observed.

PCA was first performed on GSC samples, one sample at a time to highlight intratumoral heterogeneity. A cycling-free PCA strategy (Supplementary Fig. 2a) was used since not all cells were cycling (Supplementary Fig. 2b).

For each GSC-enriched tumor sample, we found that the first principal component (PC) separated cells into neural developmental lineages. GSCs expressing neuronal genes such as CD24, SOX11, and DCX were mutually exclusive from cells expressing astrocytic (including astro-mesenchymal) genes such as GFAP, APOE, AQP4, CD44, CD9, and VIM (Fig. 1b). To assess the conservation of these gene programs across patients, we ranked genes by strength of influence on PC1 and found a strong correlation of these ranks between samples ($R^2 = 0.72$, Fig. 1c). GSCs with intermediate PC1 values express progenitor genes such as SOX4, OLIG2, and ASCL1 (Fig. 1b). In some samples, these cells had high PC2 values; however, this was not apparent in all samples and the rank correlation was lower ($R^2 = 0.37$, Supplementary Fig. 2c). We also compared the signature of each cell within the GSC-enriched samples to determine their TCGA subtype (Fig. 1b). For each patient sample, cells matching the proneural, classical, and mesenchymal signatures were present. We validated the differential gene expression profiles of enriched GSCs and whole-tumor cell populations using flow cytometry. In general, cells do not coexpress neuronal (e.g., CD24) and astrocytic (e.g., CD44) markers (Fig. 1d). Together,

**Table 1 Patient and sample information.**

| | Sample | Location | IDH mutational status | 1p19q co-deletion | ATRX | MGMT methylation | Sorting post-dissociation | Total no. of cells (after QC) | No. of tumor cells | % Tumor | Median genes per cells | Median UMI per cell |
|---|---|---|---|---|---|---|---|---|---|---|---|---|
| Whole tumor | BT333 | Right temporal, corpus callosum | WT | No | WT | Unmethylated | Yes | 852 | 614 | 72.1% | 2561 | 6800 |
| | BT338 | Right fronto-temporal | WT | No | NA | Methylated | Yes | 2131 | 1404 | 65.9% | 4227 | 14,060 |
| | BT346 | Right fronto-insular | WT | No | WT | Methylated | Yes | 2180 | 1902 | 87.2% | 3472 | 11,332 |
| | BT363 | Left frontal | WT | No | WT | Unmethylated | Yes | 10137 | 8726 | 86.1% | 2143 | 4998 |
| | BT364 | Left parieto-occipital | WT | No | NA | Methylated | Yes | 7280 | 5688 | 78.1% | 3258 | 10,305 |
| | BT368 | Left temporal | WT | No | NA | Methylated | Yes | 2647 | 2400 | 90.7% | 2904 | 8251 |
| | BT389 | Left temporo-insular | WT | No | WT | Methylated | No | 4636 | 1049 | 22.6% | 1751 | 3732 |
| | BT390 | Right temporo-occipital | WT | No | WT | Methylated | No | 3695 | 1124 | 30.4% | 2005 | 4718 |
| | BT397 | Left frontal | WT | No | WT | Unmethylated | No | 6469 | 586 | 9.1% | 2471 | 5539 |
| | BT400 | Left mesio-temporal | WT | No | WT | Unmethylated | No | 5032 | 4508 | 89.6% | 1491 | 2813 |
| | BT402 | Right fronto-parietal | WT | No | WT | Methylated | No | 4953 | 1110 | 22.4% | 2598 | 6891 |
| | BT407 | Right fronto-temporal | WT | No | WT | Unmethylated | No | 3884 | 115 | 3.0% | 2245 | 5017 |
| | BT409 | Right temporo-parieto-occipital | WT | No | WT | Methylated | No | 2470 | 979 | 39.6% | 2169 | 5016 |
| GSC | BT322 | Right temporal | WT | No | WT | Methylated | NA | 3451 | 3440 | 99.7% | 3882 | 12,568 |
| | BT324 | Left temporal | WT | No | WT | Unmethylated | NA | 5683 | 4843 | 85.2% | 2529 | 6445 |
| | BT326 | Right frontal | WT | No | WT | Unmethylated | NA | 2702 | 2692 | 99.6% | 5125 | 23,788 |
| | BT333 | Right temporal, corpus callosum | WT | No | WT | Unmethylated | NA | 5072 | 4930 | 97.2% | 3493 | 11,207 |
| | BT363 | Left frontal | WT | No | WT | Unmethylated | NA | 4919 | 4050 | 82.3% | 3410 | 9782 |
| | BT368 | Left temporal | WT | No | NA | Methylated | NA | 3470 | 3426 | 98.7% | 3099 | 8208 |

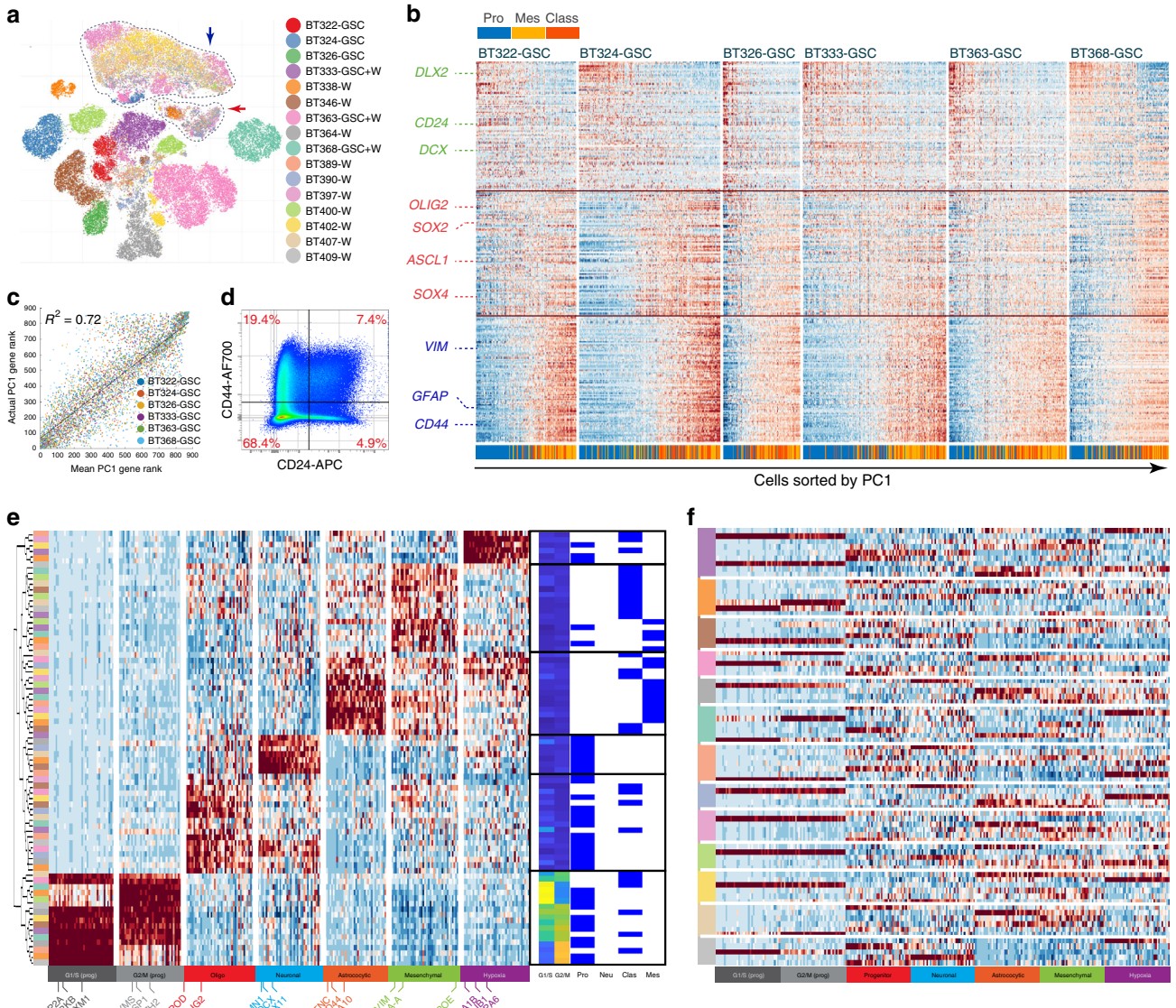

**Fig. 1 Single-cell RNA sequencing highlights transcriptomic heterogeneity in glioblastoma and glioblastoma stem cells. a** tSNE of location-averaged transcriptome for all tumor cells colored by patient. Cancer cells cluster by patient, whereas normal cells from all patients cluster together (encircled clusters indicated by arrows). GSC corresponds to glioma stem cell samples, W corresponds to whole samples. **b** Enriched glioblastoma stem cell (GSC) gene expression heatmaps showing relative gene expression (raw data) sorted by PC1 per patient. These maps are separated into three rows: top row—100 genes with the lowest value for PC1 loading; bottom row—100 genes with the highest value for PC1 loading; middle row—100 genes with the highest value for PC2 loading. These gene signatures correspond to neuronal, astrocytic, and progenitor signatures, respectively. The TCGA subtype is also shown for each GSC. **c** Mean and actual rank of genes by PC1 correlation. The actual gene rank (y axis, one point per sample) correlates strongly with the mean gene rank (x axis) in all patients. **d** Flow cytometry analysis of GSCs and whole-tumor, demonstrating mutually exclusive expression of CD24 and CD44. **e** Heatmap of gene expression by cNMF signature with associated cell cycle scores and TCGA subtype (right). The most characteristic genes for each signature group are depicted on the x axis. Signatures (y axis) are ordered according to hierarchical clustering (left tree). Left color bar represents the patient sample that generated each signature—patient colors match those in Fig. 1a. Red represents high expression; blue represents low expression. Gene signatures groupings correspond to progenitors, astro-glia (mesenchymal and classical), and neurons, with the addition of cell cycle and hypoxia signatures. cNMF—clustered non-negative matrix factorization. **f** Heatmap of gene expression by signature ordered by patient as shown by the left color bar. Genes (x axis) are in the same order as Fig. 1e. Patient colors in the color bar match those in Fig. 1a, e. Each patient contains signatures from multiple groups.

these data suggest that GSCs are organized into progenitor, neuronal, and astrocytic gene expression programs, resembling a developing brain.

We applied the same strategy to the whole-tumor samples. Once the cell cycle effect was removed (see Methods), variability in gene expression profiles remained apparent within tumors and between tumors. We identified multiple TCGA subtypes in each tumor, as previously shown[12,20] (Supplementary Fig. 2d).

Importantly, cells with different TCGA subtypes were often separated by the first of second PCs, indicating that these subtypes accurately describe a portion of the intrinsic heterogeneity of each tumor. Also, in each tumor, cells with different TCGA subtypes did not necessarily belong to different CNA clones (Supplementary Fig. 2e); however, different proportions of TCGA subtypes were observed between some clones within individual tumors. This is consistent with results from the TCGA,

indicating that genomic aberrations do not perfectly predict a subtype.

To better characterize sample heterogeneity in whole-tumor samples, we implemented a sample-wise cNMF algorithm[36]. We found five to nine signatures per sample. These clustered into seven groups (Fig. 1e and Supplementary Fig. 2f). Through identification of the most characteristic genes of each group of signatures, we found that the first and second groups expressed genes important for the G1S and G2M cell cycle programs with some stem cell genes like EZH2, whereas the seventh group expressed genes important for hypoxia response (Fig. 1e). The third and fourth groups were more closely related and expressed genes associated with oligo-progenitor and neuronal cells, respectively (Fig. 1e). The fifth and sixth groups expressed genes associated with astrocytic differentiation. Critically, each patient sample yielded signatures, which belonged to three to seven different groups (Fig. 1f).

We compared each of these signatures with those obtained in the TCGA[19] (Fig. 1e). The cell cycle, oligo-progenitor, and neuronal signatures were associated with the proneural subtype and the hypoxia signature was more associated with the mesenchymal subtype. One of the two astrocytic signature groups matched the classical subtype, whereas the other matched the mesenchymal subtype. That the classical and mesenchymal signatures clustered together and expressed astrocytic genes corroborates their resemblance to astrocytes and cultured astrocytes, respectively[19]. Finally, as was previously found by Wang et al.[20], none of the signatures matched the neural subtype.

**ScRNAseq of the normal developing brain.** If glioblastoma is organized into programs reflecting normal brain development, then a direct comparison with the developing brain at a single-cell level should provide additional insight. We performed scRNAseq on freshly isolated cells from the telencephalon of four human fetuses ranging from 13 to 21 weeks of gestation. Fluorescence-assisted cell sorting (FACS) was used to remove most microglia (CD45-positive) and endothelial cells (CD31-positive) from the samples, and to select CD133-positive cells in order to improve the resolution of progenitor and neural stem cell populations[37]. By sequencing both the total and the CD133-positive cell populations, we aimed to maintain cellular representation of development. We sequenced 12,544 cells from the total unsorted population, and 10,093 cells from the CD133-positive population.

Total and CD133-positive data sets from all fetal brains were combined in silico (Supplementary Fig. 3a) after excluding ependymal cells, and the Louvain community detection algorithm was used to group cells into cell types (Fig. 2a,b). By varying the resolution parameter of the algorithm, we chose the most stable clustering solution (Fig. 2b and Supplementary Fig. 3b). This generated a total of 10 cell clusters (Fig. 2a). Differential gene expression analysis of these clusters (Supplementary Data 1) identified important genes per cluster. Cluster names were given based on their correlation with cell types described by Nowakowski et al.[38] (Supplementary Fig. 3c, d). CD133-positive cells were found in all clusters/cell types, but were enriched in the radial glia, neuronal progenitors, and committed glial cell clusters (Fig. 2d and Supplementary Fig. 3e).

Two CD133-positive cell types did not fit with previously identified gene signatures[38]. The first was detected mainly in the 17- and 19-week brains and highly expressed genes such as VIM, GFAP, OLIG1, GLI3, and EOMES (Fig. 2a, uRG). It was found to be most similar to certain types of radial glia in the Nowakowski et al. data set[38] (Supplementary Fig. 3c, d). The second cell type, an unidentified glial cell cluster, was detected at all gestational ages and strongly expressed oligodendrocyte lineage

genes (e.g., OLIG1, OLIG2, and PDGFRA), glial/astrocytic lineage genes (e.g., GFAP, SOX9, HOPX, HEPACAM, and VIM), and progenitor genes (e.g., ASCL1, MKI67, and HES6) (Fig. 2c, dotted line encircles the glial cluster as well as oligodendrocytic cells and astrocytes for comparison, and Supplementary Data 1). Accordingly, high correlation was observed with astrocytes and oligodendrocyte progenitor cells in the Nowakowski et al.[38] data set (Supplementary Fig. 3c). However, it did not express differentiation markers found in astrocytes or oligo-lineage cells (OLCs) such as APOE and APOD, respectively (Fig. 2c). It also lacked the high gene complexity and UMI counts seen in doublets (Supplementary Fig. 3f). This mixed gene signature is compatible with that of a bipotential glial progenitor cell (GPC). Notably, this GPC signature was almost exclusively identified in CD133-sorted cells (Fig. 2a, d and Supplementary Fig. 3e), which likely explains why it was not previously detected[38,39]. The existence of cells expressing these GPC markers was confirmed in first passage culture of fetal brain cells derived from one of the fetal brains sequenced (Fig. 2e), and in the subventricular zone of the adult human brain (Fig. 2f).

**Creation of a fetal brain roadmap.** We next aimed to find a parallel for each cancer cell to a fetal brain cell type. To do so, we developed a roadmap technique that enables the projection of every cancer cell onto the fetal data set. We first selected the appropriate fetal cell types to build the roadmap. This was accomplished by determining which fetal brain cell type was nearest to, or captured, each cancer cell. Ninety-four percent of whole-tumor cells were captured by five fetal brain cell types: neurons; astrocytes; OLCs; truncated radial glia (tRG); and GPCs (Supplementary Fig. 4a). Surprisingly, interneurons captured more cells than excitatory neurons. Consequently, the five cell types used to construct the roadmap were astrocytes; tRG; GPCs; OLCs; and interneurons.

We used PCA on an equal number of fetal astrocytes, GPCs, OLCs, interneurons, and tRG. This fetal PC space acts as the roadmap. We then used diffusion embedding[40,41] to better represent the differentiation process in 3D. In this diffusion roadmap, GPCs are found at the junction of the oligodendrocytic, astrocytic, tRG, and neuronal lineages (Fig. 3a).

**Fetal brain roadmap reveals glioblastoma trilineage hierarchy.** We projected an equal number of cancer cells from each patient onto this roadmap and used the first three components of diffusion embedding which most effectively separated the cell types as each cell's coordinate in the hierarchy (Fig. 3b), with the exception of one tumor from which we obtained too few cells (BT407-W). GSCs and whole-tumor cells overlapped (Fig. 3c) despite significant variations in lineage proportions between patients in the whole-tumor samples (Supplementary Fig. 4c). Mirroring the TCGA analysis performed above (Supplementary Fig. 2e), different cell types did not necessarily belong to different CNA clones (Supplementary Fig. 4c). The GPC signature was the only one robustly expressed in all patients. To visualize gene signatures, we ordered cancer cells according to all three diffusion components (DCs) individually and found genes that correlated most with this order (Fig. 3d and Supplementary Data 2). Cancer cells expressing an OLC signature (e.g., OLIG1, APOD; Supplementary Fig. 4d) or astro-mesenchymal signatures (e.g., CD44, GFAP, AQP4) were found at either end of DC2; cancer cells expressing a neuronal signature (e.g., STMN2, DLX2) were found at the end of DC1; and cancer cells expressing a GPC signature (e.g., OLIG2, NES ASCL1, HES6) were found at the end of DC3 and mid-DC2. We therefore defined DC3 as the glial progenitor

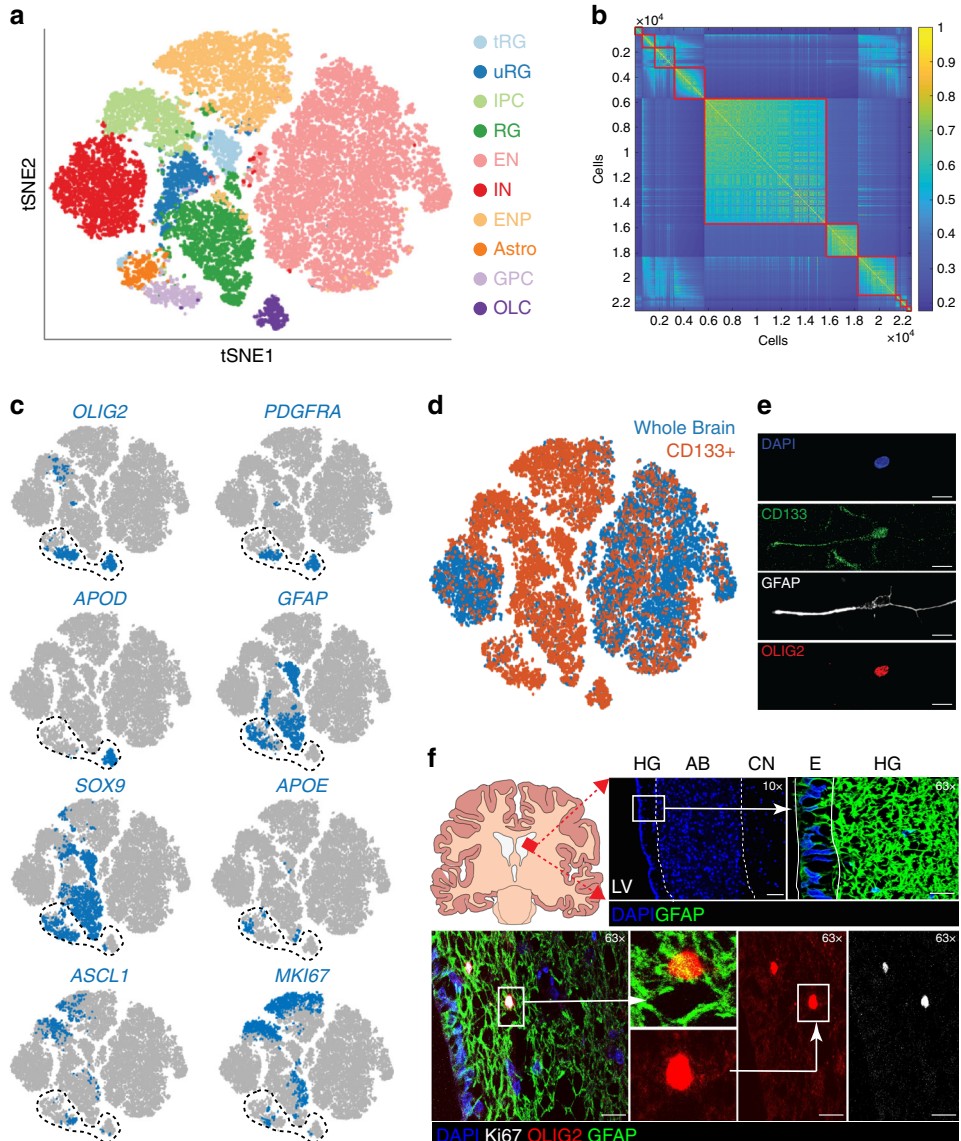

**Fig. 2 Single-cell RNA sequencing of the developing brain and the identification of glial progenitor cells. a** T-distributed stochastic neighbour embedding (tSNE) map of human fetal brain cells by cluster or cell type. Data sets from total cells and CD133+ cells were combined. Cells are colored by cell type. *tRG* truncated radial glia, *uRG* unknown radial glia, *IPC* inhibitory neuronal progenitor, *RG* radial glia, *EN* excitatory neuron, *IN* interneuron, *ENP* excitatory neuronal progenitor, *Astro* astrocyte, *GPC* glial progenitor cell, *OLC* oligo-lineage cells. **b** Similarity matrix of fetal brain cells ordered by cluster. **c** tSNE maps of human fetal brain cells showing cell type expression of OLIG2, PDGFRA, APOD, GFAP, SOX9, APOE, ASCL1, and MKI67. Expression is averaged to the 20 closest neighbors in principal component (PC) space. Encircled cells were reclustered to yield three separate clusters. **d** tSNE map of total human fetal brain cells and CD133+ fetal brain cells. **e** Representative example of freshly cultured fetal neural stem cells coexpressing CD133, OLIG2, and GFAP (*n* = 2 independent biological samples). Images were taken at ×63 magnification. Scale bars: 10 μm **f** Immunofluorescence analysis of the adult human subventricular zone (SVZ) at the junction of the AB and HG. Top row, schematic and anatomic structure of the SVZ. Bottom row, identification of dividing cells with marker expression corresponding to glial progenitor cells. *HG* hypocellular gap, *AB* astrocytic band, *E* ependymal cells, *LV* lateral ventricle, *CN* caudate nucleus. Analysis was performed in *n* = 4 independent patient samples. Scale bars: top row images: 200 μm (left) and 40 μm (right); bottom row images: 20 μm.

score. This organization reveals a glial progenitor-centered trilineage organization of whole-tumor and enriched GSCs.

When comparing enriched GSCs and whole-tumor cells, we found a significant shift of GSCs toward higher values on the glial progenitor score ($p < 1E-21$), and a shift toward intermediate values of DC2 (Fig. 3c and Supplementary Fig. 4e). These data show that glial progenitor cancer cells are enriched in GSC culture conditions, and a shift away from the astrocytic, mesenchymal, and oligodendrocytic cancer cell types occurs after 7 days in stem cell culture conditions.

Lastly, cancer cells from whole tumor were classified into cell types using a LDA with the fetal cells as a training set (Fig. 3a, b). Cells that could not be classified with a probability of error <0.01% were left unclassified (Fig. 3b); these correspond to cells with intermediate signatures. The gene expression profile of the roadmap cell types closely matched those obtained by cNMF (Supplementary Fig. 4f). We also found a close agreement between these cell type signatures and the signatures described by Neftel et al.[42], highlighting the fundamental nature of these lineages in IDHwt glioblastoma (Supplementary Fig. 4f).

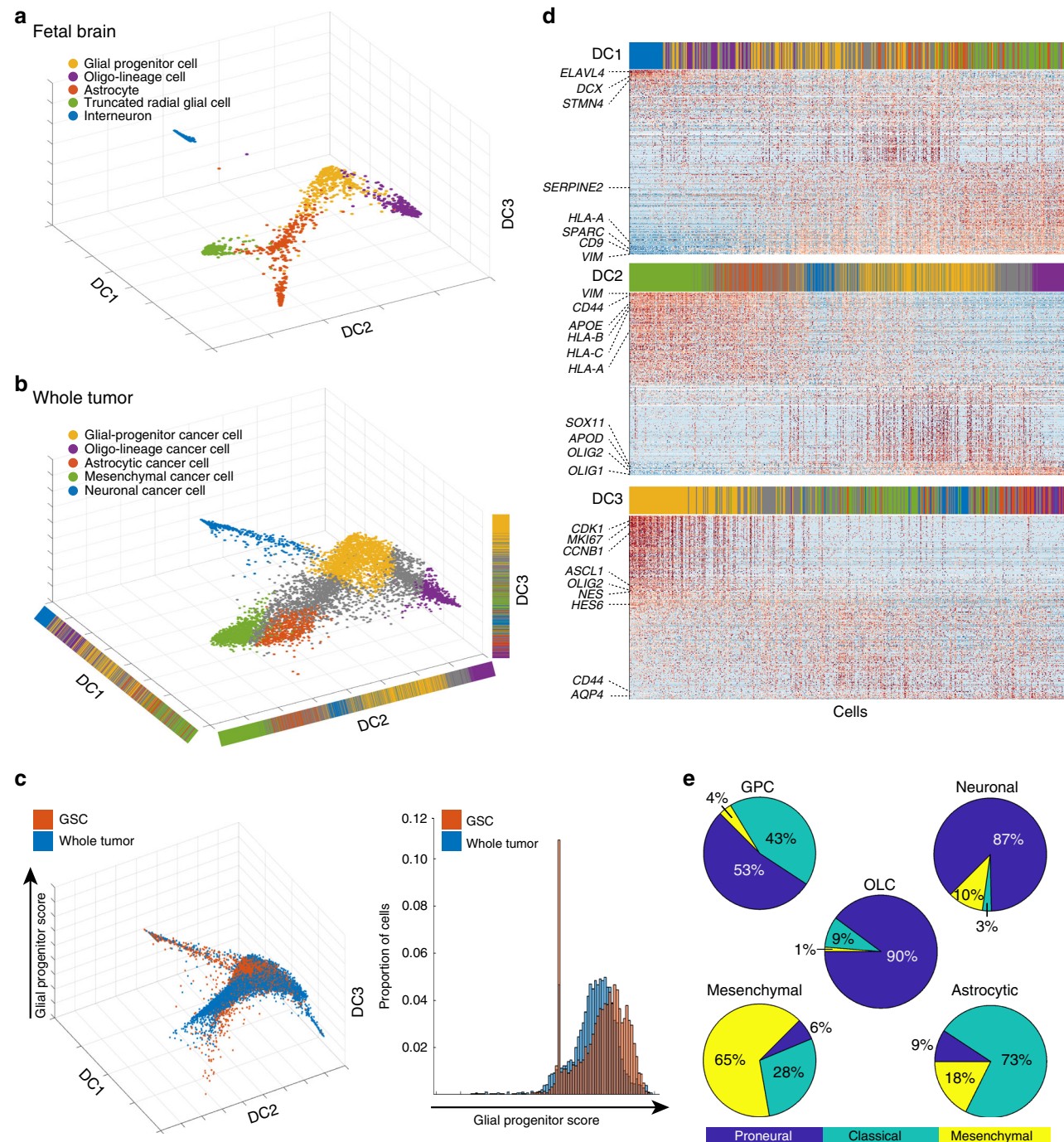

**Fig. 3 Fetal brain roadmap reveals a glioblastoma trilineage hierarchy centered on progenitor cancer cells. a** Diffusion plot of the projection of selected fetal cell types onto the roadmap. Cells are colored by the cell type they were attributed in Fig. 2a. **b** Diffusion plot of the projection of an equal number of whole-tumor cancer cells from each patient onto the roadmap. Cells are colored based on their classification by linear discriminant analysis (LDA). Unclassified cells were colored gray. **c** Diffusion plot showing the location of glioma stem cells (GSCs) relative to whole-tumor cells (left) and histogram of glial progenitor score for GSCs and whole-tumor cells (right). An increase in proportion of cells with higher glial progenitor scores is seen in GSCs ($p < 1e$-21, two-sample Kolmogorov–Smirnov test). Only samples with paired GSC and whole-tumor data were used here. **d** Heatmaps showing relative gene expression (raw data) for cells ordered by each of the diffusion components of the roadmap. Genes are ordered from most correlated to least correlated with the diffusion component. The 200 most and 200 least correlated genes are shown. Top color bar indicates cell type classification from the LDA. Each color corresponds to the same classification as in **b**. **e** Pie chart for TCGA subtype by cell type for a subset of 1000 cells. Cell types are based on the LDA classification for all whole-tumor cells. and TCGA subtype was obtained using Gliovis (see Methods).

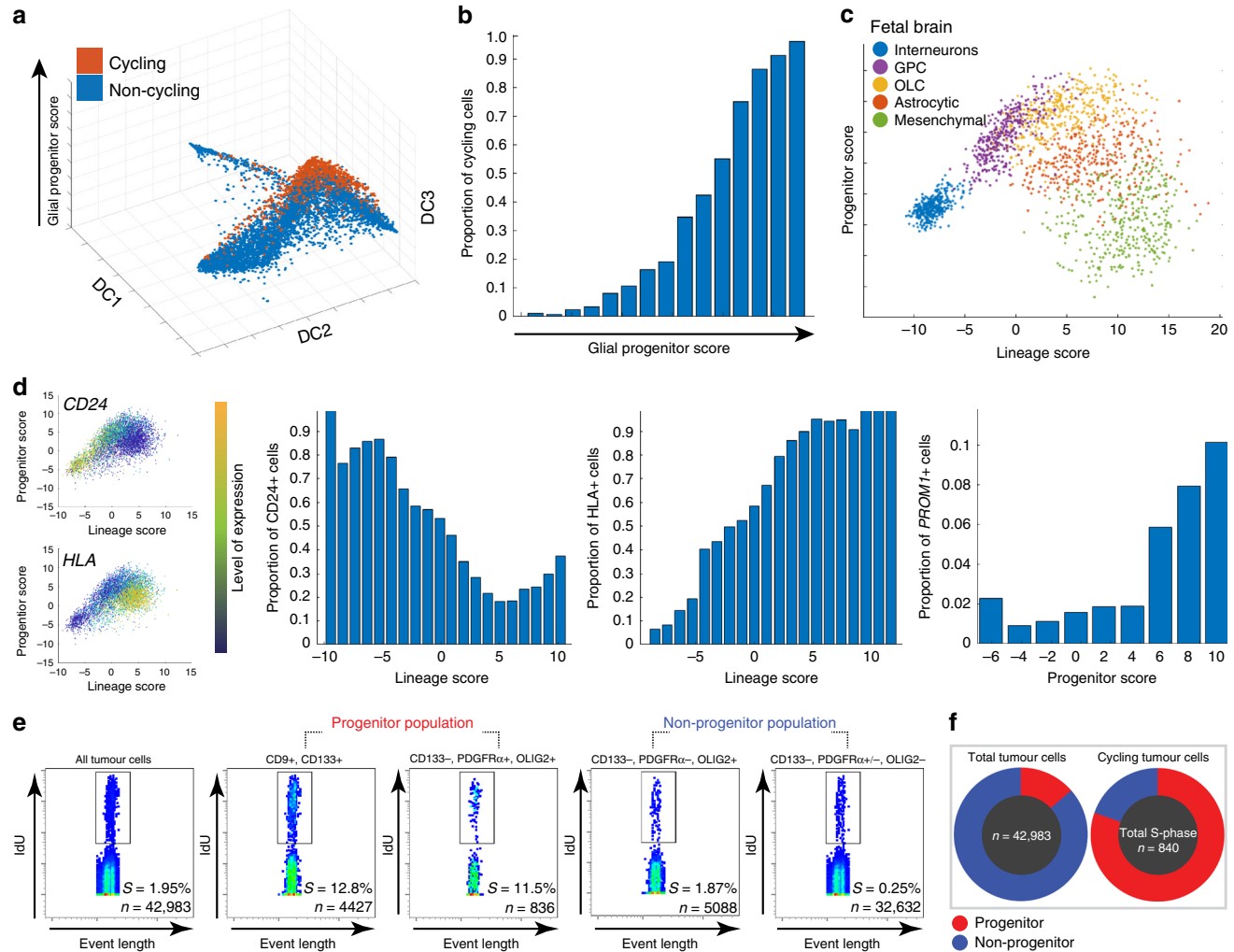

**Fig. 4 Progenitor cancer cells are the most proliferative cancer cells. a** Diffusion plot of the roadmap of whole-tumor cancer cells showing that cycling cells are predominantly glial progenitor cancer cells. Cycling cells are defined by >1.5 in either the G1/S or G2/M scores. **b** Bar chart showing that the proportion of cycling cells increases with increasing glial progenitor score. **c** Simplified roadmap in principal component space with select fetal brain cell types. Projected OLCs and GPCs overlap and are high for progenitor score, whereas interneurons, and tRG/astrocytes are lower in progenitor score, but occupy opposite ends of the lineage score. **d** Projection of glioma stem cells (GSCs) on the simplified roadmap highlights the location of CD24 and HLA within the hierarchy. For each gene, the simplified roadmap projection shows the expression of this gene in GSCs, and the histograms show the proportion of cells where CD24, HLA, and PROM1 (CD133) were detected at differing positions in the hierarchy. **e** Mass cytometry pseudo-color dot-plots showing the proportion of whole-tumor cells, progenitor cancer cells, and non-progenitor cells that are in S-phase. The progenitor cancer cell population has the highest proportion of cells in S-phase (box). **f** Mass cytometry showing that progenitor cancer cells are the main cycling cell population in the tumor. Pie charts showing the proportion of progenitor cells (CD133+, OLIG2+, PDGFRA+) in the tumor (left) and the cycling population (right).

We compared the TCGA subtype of each cell (see TCGA analysis above) with its classified cell type (Fig. 3e). As predicted by previous work[19,42], neuronal and oligo-lineage cancer cells were almost exclusively proneural; astrocytic cancer cells were strongly classical; and mesenchymal cancer cells were strongly mesenchymal. Glial progenitor cancer cells were mostly proneural (Fig. 3e), but similarities to the classical and mesenchymal subtypes were also found. We then used our signatures to score the TCGA samples[19] according to these cell type signatures (Supplementary Fig. 4g). The proportion of cell types by TCGA subtype is in close agreement to that described by Neftel et al.[42]

**Progenitor cancer cells are the most proliferative cancer cells.** Based on the expression of cell cycle genes, we defined a cycling cell as one with a G1/S or G2/M score >1.5, as was done previously[25]. We then calculated the proportion of cycling cells as a function of their glial progenitor score. We found almost all cycling cancer cells had high glial progenitor scores (Fig. 4a, b).

We then aimed to validate this result using single-cell proteomic analysis. To do so, we generated protein marker panels representative of each cancer cell type. We created a simplified roadmap (see methods) with progenitor and lineage scores (Fig. 4c). We projected GSCs onto this modified roadmap and selected genes encoding cell surface protein markers which most strongly correlated with the lineage scores (Fig. 4d and Supplementary Data 3). Interestingly, this projection high-lighted the relative absence of HLA gene expression in neuronal cancer cells (Fig. 4d), analogous to normal neurons[43]. This may have implications in the immune responsiveness of these cells. For the purposes of cytometry assays and sorting, we defined CCD133−/CD24+/CD9− as neuronal cancer cells, CD9+/CD44+/CD133− as astro-mesenchymal cancer cells, and CD9+/CD133+ as progenitor cancer cells. For the mass

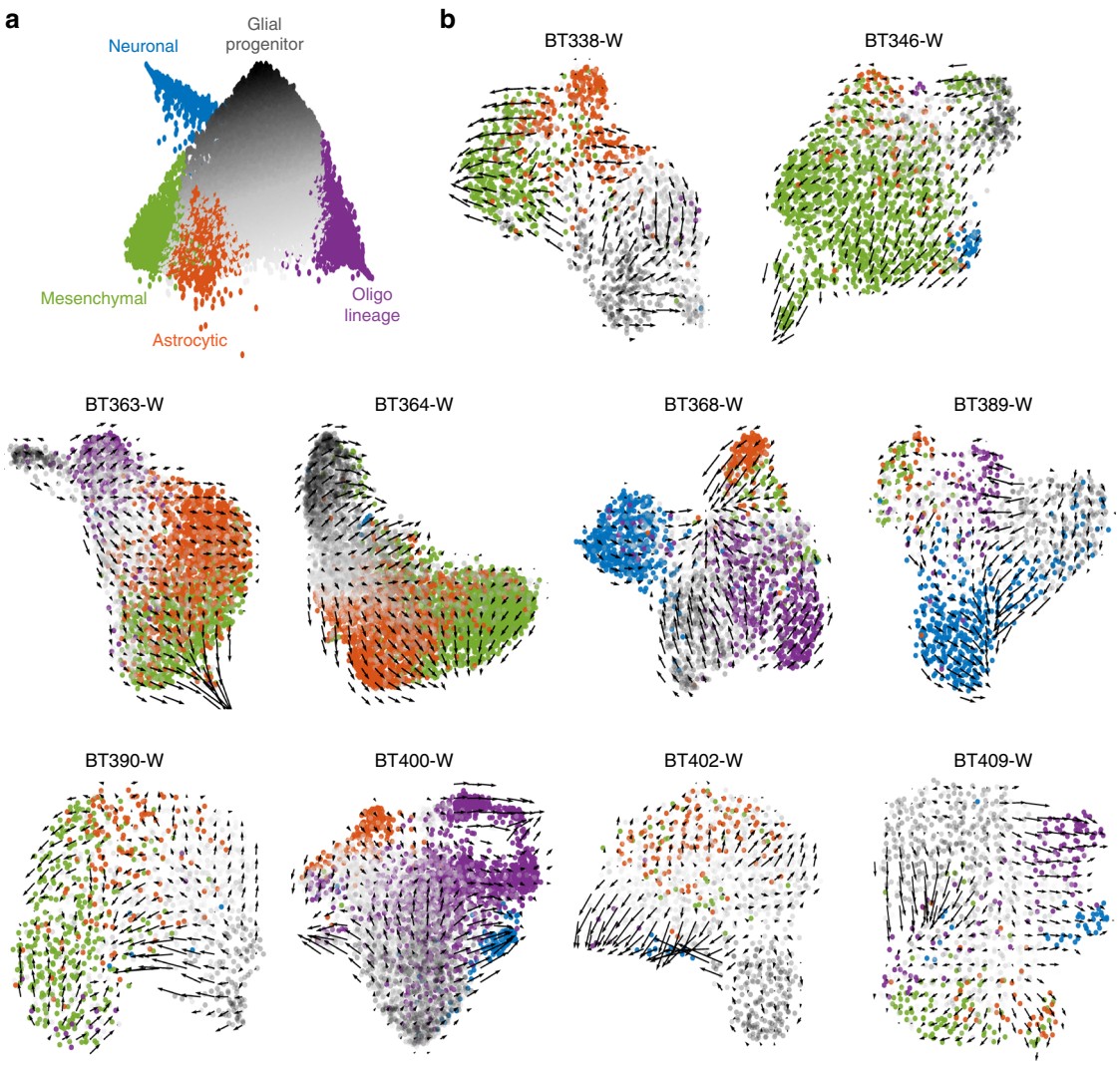

**Fig. 5 RNA velocity supports conserved hierarchical dynamics in glioblastoma. a** Diffusion roadmap schematic for all whole-tumor samples where progenitor cells and unclassified cells were colored according to a grayscale: cells scoring higher on the progenitor axis are darker. **b** Velocity field superimposed to the UMAP embedding of cells by sample. Cells are colored by cell type according to **a**. *UMAP* uniform manifold approximation and projection.

cytometry assay, PDGFRA and OLIG2 were included as markers for progenitor cancer cells.

Using the progenitor cancer cell marker panel, and a validated cell cycle marker panel[44,45], we used mass cytometry to analyze 42,983 cancer cells, and found 840 cells in S-phase (1.95%) (Fig. 4e). We found that 12.6% of the progenitor cancer cell population was in the S-phase. These progenitor cancer cells made up only 12.2% of the total tumor population yet accounted for 78.9% of all S-phase cells (Fig. 4f). Interestingly, much of the remaining S-phase cells expressed a subset of the progenitor signature (Fig. 4e), highlighting the continuous nature of differentiation. In contrast, only 0.25% of cells without progenitor markers were found in S-phase (Fig. 4e). Together, these non-progenitor cells made up 87.8% of the total population but only 21.1% of all cells in S-phase (Fig. 4f). Similarly, tumor immunolabeling, using Ki67 as a marker of cell proliferation, showed that the percentage of cycling cells in the CD133-positive population is significantly higher than that of CD133-negative population in two patients (Supplementary Fig. 4h).

**Progenitor cancer cells at apex of glioblastoma hierarchy.** We found that all samples had high intron rates similar to those

observed in mouse brain development[46] (Supplementary Fig. 5a). Therefore, we used RNA velocity[46] to measure transcriptional dynamics and characterize the differentiation process in glioblastoma.

To find patterns in velocities, we labeled cells with the cell type classification they were given in the LDA described above. Progenitor cells and unclassified cells were colored according to a greyscale, which indicated the magnitude of their progenitor score (Fig. 5a), highlighting the spectrum of differentiation seen in the roadmap. Notably, cells with the same cell type aggregated together and were at the periphery of the UMAP (Fig. 5b), suggesting once again these cell types are intrinsic to the glioblastoma samples.

Directional flow was noticed in every patient sample (Fig. 5b). We confirmed this was not owing to random chance (representative example in Supplementary Fig. 5b). In general, the vector field points from cells with high glial progenitor scores to cells classified to a specific lineage (Fig. 5b). We also performed velocity with PCA embedding, a mathematically simpler representation than UMAP. These data also show that the main direction of flow is from progenitor cells to differentiated cell types (Supplementary Fig. 5c).

Although in a few patients no clear path could be found leading to astrocytic and/or neuronal lineages, we found no clear vector paths between lineages. An exception to this is the mesenchymal cell type, which appeared downstream of astrocytic cancer cells in most patient samples (Fig. 5b), or appeared to be intermediate between progenitors and other lineages in one patient sample (BT346). Interestingly, in samples containing multiple cell types (e.g., BT389, BT400, and BT409), cells often did not completely segregate by lineage until the terminus of their respective lineage velocity field.

Together, these analyses suggest that astrocytic, mesenchymal, oligodendrocytic, and neuronal cancer cells are more differentiated than progenitor cancer cells, and that the latter are most often the originator of the hierarchy in glioblastoma.

**Progenitor cancer cells drive chemoresistance and growth**. Resistance to conventional chemotherapies and tumorigenicity are hallmarks of CSCs[6,17,18]. These data, however, are derived from studies that have considered the CSC compartment to be uniform, not one displaying heterogeneity driven by a hierarchical developmental organization. To evaluate GSC chemoresistance and tumorigenicity considering hierarchy and lineage, we sorted them into following three types: progenitor, neuronal, and astro-mesenchymal, based on the protein expression panel described above.

Three patient-derived GSC lines were separated into these types and treated with TMZ. Variable doses were required to achieve responses in different cell lines, correlating with the methylguanine methyltransferase status of the tumor. We found that progenitor GSCs either did not respond or responded less to TMZ than the more-differentiated GSCs (Fig. 6a and Supplementary Fig. 6a).

We then assessed the influence of hierarchy and lineage on tumor forming capacity. Forty-seven mice were orthotopically xenografted with progenitor, neuronal, astrocytic, or total GSCs from three different patients in near-limiting dilution. We observed earlier tumor formation, and a more rapid increase in tumor signal, for all mice implanted with progenitor or total GSCs. In mice implanted with astro-mesenchymal or neuronal GSCs the tumor formation and signal increase was either absent or significantly delayed by up to 3 months (Fig. 6b–d). Consistently, mice implanted with progenitor GSCs had a significantly lower survival time than those implanted with neuronal (OR 0.26, $p < 0.01$) or astro-mesenchymal (OR 0.05, $p < 0.001$) GSCs (Fig. 6e).

We also analyzed the xenografts to determine the progeny of each implanted GSC cell type (Fig. 6d). At 12 weeks, progenitor GSCs gave rise to tumors expressing mainly the progenitor marker ASCL1 and small populations of cancer cells expressing the neuronal marker DCX or the astro-mesenchymal marker CD44. Neuronal GSCs gave rise to tumors expressing mainly DCX, and smaller populations of cells expressing ASCL1. No CD44-expressing cells were found in these tumors. Finally, the very small tumors stemming from astro-mesenchymal GSCs expressed mainly ASCL1 and a small population of CD44-expressing cancer cells. The low proportion of CD44-expressing cells may be due to the lack of immune micro-environment in NSG mice[47].

Together, these results identify a lineage hierarchy of tumorigenicity and chemoresistance in GSCs, with progenitor cancer cells being the most chemoresistant and tumorigenic. Our findings also indicate that lineage specificity and plasticity exist within the GSC pool.

**Progenitor pathways expose therapeutic opportunities**. As progenitor GSCs are the most chemoresistant and tumorigenic cancer cell population, we aimed to leverage our hierarchy and

transcriptomic data to find targets relevant to this cancer cell population.

We used the LDA classification of whole-tumor cells described above to separate cells into cell types. We selected the GPC and astro-mesenchymal groups for the analysis to specifically compare the progenitor population to the most abundant cell types in the cancer. We performed gene set enrichment analysis (GSEA) in a manner similar to previously described methodologies[48]. We identified pathways with a significant enrichment in progenitor cancer cells (Supplementary Data 4). Hits with significant and strong correlations were found in pathways such as EZH2, FOXM1, and Wnt, previously established pathways relevant to CSC self-renewal and tumorigenicity[49–51].

Pathways of previously unknown significance in GSCs were also detected. Of these, the E2F4 pathway was the most significant, and it was thus selected to test our target identification method. The E2F gene family regulates cell cycle and is important for progenitor cell survival[52]. The E2F4 gene set involves many of the regulating targets of the transcription factor E2F4; therefore, E2F4 inhibition was selected to target this pathway. HLM006474 is a small molecule inhibitor that prevents E2F4 binding to DNA. It has been shown to cause senescence of gastric cancer cells[53], and to reduce proliferation and survival of melanocytic cells and lung cancer cells in vitro[54,55]. E2F4 expression in glioblastoma tissue has been shown[56]. To our knowledge, our work provides the first description of its importance in GSCs.

We tested the effect of E2F4 inhibition in progenitor, neuronal, and astro-mesenchymal GSCs following HLM006474 treatment. Proliferation and survival of progenitor GSCs was significantly reduced compared with neuronal and astro-mesenchymal GSCs (Fig. 7a). This differential sensitivity was also observed in a sphere forming capacity assay (Fig. 7b, c) and serum-free vs serum-differentiated GSCs (Supplementary Fig. 6b). On target E2F inhibition was confirmed[54] (Supplementary Fig. 6c, d). Together, these data show that targeting E2F4 preferentially affects progenitor GSC proliferation.

We tested the effects of E2F4 inhibition in vivo. Pooled GSCs, treated with HLM006474 or vehicle for 3 days, were orthotopically xenografted. A significant reduction in tumor growth (Fig. 7d, e), and improved survival (Fig. 7f, $p$ value $= 0.03$, Cox proportional hazard) in the HLM006474-treated mice was observed.

As E2F4 inhibition is effective in progenitor GSCs, and TMZ chemotherapy is more effective in more-differentiated GSCs, we reasoned that HLM006474 combined with TMZ would be a more-effective treatment for the total GSC compartment than each individually. We sequentially treated GSCs with HLM006474 followed by TMZ chemotherapy at TMZ doses that are ineffective as monotherapy. We observed a further decrease in proliferation and cell survival using this combination therapy compared with monotherapy (Fig. 7g). We performed an isobolographic analysis of this combination therapy[57] to assess for synergism or antagonism. We found no significant difference between the measured isoboles and the control additive isobole (Supplementary Fig. 6e, $p$ value $= 0.74$, Student's $t$). Therefore, no synergism or antagonistic effect was found between the two compounds, but their additive properties suggest they could be used in combination to treat both progenitor and more-differentiated GSCs within the total GSC population.

## Discussion

Intratumoral and interpatient heterogeneity are hallmarks of many cancers[1]. Here, we show that the normal developing human brain can be used as a roadmap to elucidate brain cancer development, and, in conjunction with RNA velocity, reveal that

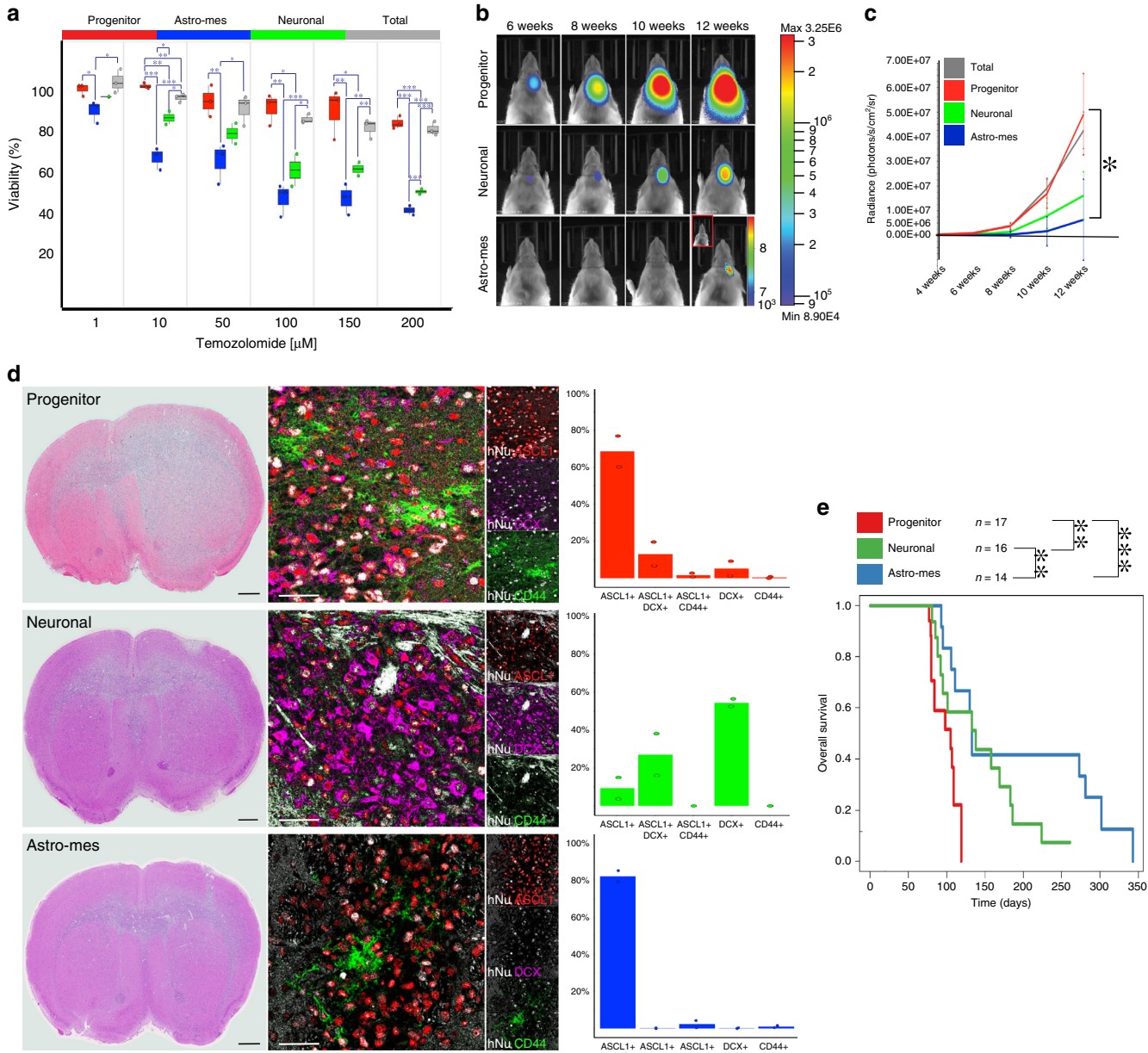

**Fig. 6 Progenitor cancer cells are drivers of chemoresistance and tumor growth. a** Box–whisker plots showing the proportion of viable glioma stem cells (GSCs, *n* = 1 patient: BT390-GSC) sorted by type and followed by 5 days of temozolomide (TMZ) treatment, normalized to corresponding vehicle control. See Supplementary Fig. 5e for additional patients. Three technical replicates and three biological replicates were performed per condition. Box plot represents the first quartile, median, and third quartile with whiskers corresponding to 1.5 times the interquartile range. The overlaid dot-plots represent the mean value per biological replicate per group. A one-tailed, two-sample equal variance *t* test was used. **b** Select bioluminescence images from mice implanted with GSCs sorted by type. Mice implanted with progenitor GSCs exhibit a more rapid tumor growth compared with those implanted with neuronal or astrocytic GSCs. **c** Average bioluminescence intensity over time for mice xenografts injected with different GSC types sorted from BT333-GSC (*n* = 24). Data are represented as mean ± SE. *p* values obtained with two-tailed, two-sample *t* tests. **d** Mice from each GSC group was killed at 12 weeks and the corresponding H&E and immunofluorescence images for cell type markers are shown. Expression of cell type-specific markers was quantified from ~1000 to ~3000 human nucleoli (hNu)-positive cells per mouse model group. Each graph represents *n* = 2 biologically independent mouse brain sections. Scale bars: whole mount images: 1 mm; immunofluorescence images: 50 μm. **e** Kaplan–Meier survival curves for mice implanted with different GSC types (*n* = 47). Univariate Cox proportional Hazard Model (two-sided) shows a significant difference in survival between progenitor GSC and neuronal (*p* value = 0.0025) or astrocytic GSC (*p* value = 5.7e-6) xenografts, and also between neuronal and astrocytic GSC xenografts (*p* value = 0.0059). For all plots, \*\*\**p* < 0.001, \*\**p* < 0.01, \**p* < 0.05.

glioblastoma develops along conserved neurodevelopmental gene programs and contains a rapidly dividing progenitor population. These data shed new light on IDHwt glioblastoma CSC hierarchy and the origins of heterogeneity.

Recently, scRNAseq characterization of human fetal brain cells described the transcriptomic signature of many cell types within

the developing brain[38,39]. By increasing the number of cells sequenced, and enriching for neural stem cells, we uncovered a cell type with a transcriptomic signature suggestive of a GPC. Additional work such as fate mapping will be necessary to uncover the exact position of these cells within the developmental hierarchy of the brain.

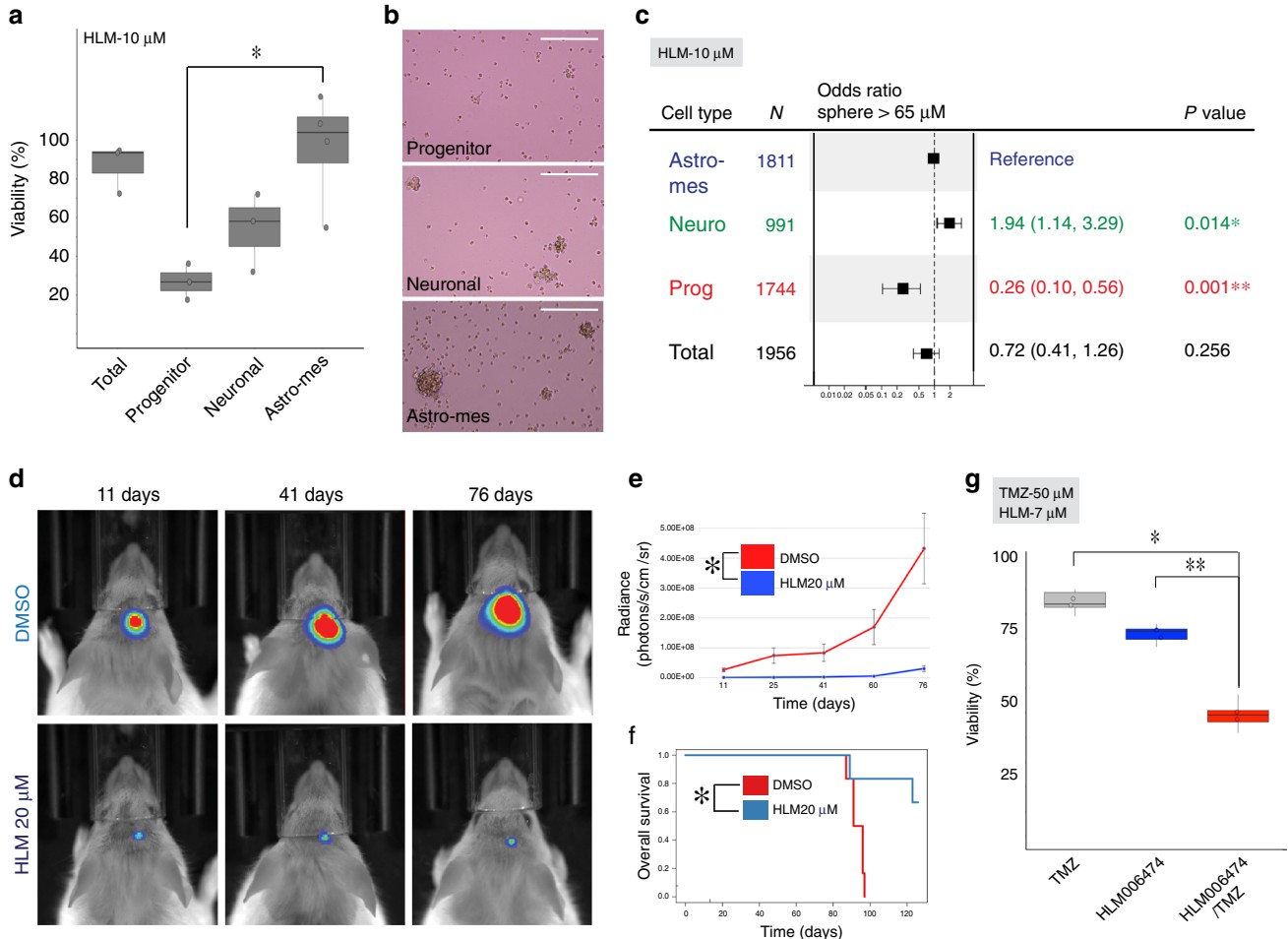

**Fig. 7 Pathways enriched in progenitor cancer cells expose therapeutic opportunities. a** Bar-graph showing the proportion of viable glioma stem cells (GSCs, BT333-GSC) sorted by type followed by 7 days of HLM006474 treatment, with each cluster normalized to corresponding vehicle control. Each bar in the graph represents the average of *n* = 3 biological replicates treated with HLM006474 as a ratio of the average of *n* = 3 DMSO-treated biological replicates. Data represented as mean±SE. A one-tailed, two-sample equal variance *t* test was used. **b** Representative images of GSCs at ×10 magnification (brightfield), sorted by type and treated in HLM006474 for 7 days (images correspond to 7a). *n* = 2 biologically independent sphere forming experiments were performed. Scale bar: 400 μM. **c** Forest plot showing the odds ratio of forming a tumor sphere >65 mm following 7 days of HLM006474 treatment, calculated using a multivariate logistic regression with the astrocytic GSC type as a reference, controlled for patient cell line. There was no significant difference between the two GSC lines (*p* > 0.2), odds ratio with 95% confidence intervals are shown. **d** Bioluminescence images and **e** signals from representative mice treated with 20 mM HLM006474 vs DMSO with corresponding **f** Kaplan−Meier survival plot (*n* = 16, eight per group). Data are represented as mean±SE. **g** Box−whisker plot showing the proportion of viable unsorted GSCs (BT326-GSC) after one of the following treatments: 6 days of TMZ treatment, 6 days of HLM006474 treatment, or 3 days of HLM006474 treatment followed by 3 days of TMZ treatment, normalized to corresponding vehicle control. *n* = 2 biological replicates per treatment. Box plot represents the first quartile, median, and third quartile with whiskers extending to 1.5 times the interquartile range. The overlaid filled dot-plots represent the mean value per biological replicate per group. *P* values: TMZ/HLM, 0.004; HLM/combination, 0.01. For all plots, **\**p* < 0.01, *\*p* < 0.05.

We showed that IDHwt glioblastoma is hierarchically organized into three cell lineages that correspond to all three normal neural lineages: astrocytic; neuronal; and oligodendrocytic. Interestingly, the neuronal lineage is devoid of HLA expression, suggesting a potential source of resistance to immunotherapy. We identified a fourth cell type by cNMF, mesenchymal, which most closely resembles tRG in our roadmap. Although this resemblance served the purpose of properly separating cancer cell types, mesenchymal cancer cells lack expression of important tRG genes such as AQP4, FAM107A, SOX9, and GLI3, and tRG lack the expression of mesenchymal genes such as CD44 and TIMP1. Therefore, tRG may not be a perfect parallel for mesenchymal cancer cells, and this parallel may not exist in the normal brain. These four cancer cell lineages closely resemble the signatures recently described by Neftel et al.[42], showing their fundamental importance in describing cancer heterogeneity.

Critically, we found a fifth cell type, which exists at the intersection of these lineages and corresponds transcriptomically to progenitors and functionally to apical glioma stem cells. The closest transcriptomic parallel of this cell cluster in the normal developing human brain are GPCs. We propose that the cell of origin of glioblastoma, whether a GPC or another cell type nearby in the brain stem cell hierarchy, also possesses such pluripotency. A genetic mouse model studying glioma origin suggested that OPCs are candidate cells of origin[58]. These cells expressed Pdgfrα, Olig2, and occasionally nestin. In our data set, both GPCs and OPCs express PDGFRA and OLIG2, but NESTIN expression is restricted to GPCs. In contrast, scRNAseq studies of IDH mutant gliomas identified only two lineages, astrocytic and oligodendrocytic[24,25]. This suggests a different cell of origin in these pathologies than in adult IDHwt glioblastoma and may underlie the disparate natural histories and treatment responses between these cancer types.

Our data also show that progenitor cancer cells are the cancer cell type with the highest rates of proliferation, more so than cancer cells undergoing lineage differentiation. The identification of highly proliferative apical CSCs here is in contrast to the work of Patel et al.[12] and Neftel et al.[42], where such a progenitor cell population was not identified. As genetic anomalies are most often acquired during the cell cycle, it is likely that new clones arise within this progenitor population and propagate down the lineages as their progeny differentiate. We suspect that specific genomic anomalies skew differentiation towards one lineage or another, giving rise to the observed TCGA subtypes. In support of this hypothesis, Neftel et al.[42] recently showed how some anomalies are associated with particular cancer cell types, and, in a model system, that these anomalies can influence the differentiation of normal neural stem cells.

These results also provide insight into hierarchy and plasticity within glioblastoma. With our high cell numbers, we estimated RNA velocity in the cancer, as was previously done in the developing mouse brain[46]. This analysis suggests progenitor cancer cells have the potential to differentiate into all cancer cell lineages identified. Although the velocity data demonstrate the main flow of differentiation, from progenitor cancer cells to the more-differentiated cell lineages, the apparent proximity of some lineages close to the progenitor population suggests that plasticity[42,59] can occur, particularly in the GSC population. In support, our GSC cell type-specific xenograft models show evidence of both lineage commitment and plasticity within the GSC population. In a similar type of experiment, Neftel et al.[42] recently used barcoded glioma cells grown in serum-free media supplemented with EGF and FGF to show that many of these cells can ultimately give rise to multiple lineages in a mature xenograft.

Our findings suggest that progenitor cancer cells are the most common originator of the cancer cell hierarchy, are the most rapidly cycling cancer cell type, and within the GSC pool are the most tumorigenic in xenograft models, more so than distal CSCs, thus driving cancer growth. In GSC culture conditions where all GSCs retain the ability to divide, progenitor GSCs are also most resistant to TMZ. Together, these findings are relevant to cancer biology and therapeutics development. These rapidly cycling progenitor cancer cells are often at apex of the cancer cell hierarchy and thus serve as a prime cell population to target. Identification of mutations or driver events within this cell population, or the identification of signaling pathway alterations between progenitor cancer cells and more-differentiated cancer cells will likely yield meaningful new therapeutic targets.

To that end, we leveraged our transcriptomic data and conserved hierarchical neurodevelopmental classification to identify therapeutic targets relevant to progenitor cancer cells in all patients. HLM006474, an E2F4 blocker, shows pronounced activity toward progenitor GSCs versus GSCs that have differentiated towards the neuronal or astrocytic-mesenchymal lineages. We showed that E2F4 inhibition significantly hampered tumor growth in vivo. As mice xenografted with these progenitor GSCs develop tumors faster and exhibit a shorter survival time than mice engrafted with distal GSCs, targeting this most rapidly cycling and functionally aggressive progenitor cancer cell population may be an effective treatment approach. Given the plasticity that can occur in the GSC population, separate targeting of all cell types within the cancer will likely be needed.

## Methods

**Glioblastoma samples**. Glioblastoma samples were harvested under a protocol approved by the Montreal Neurological Hospital's research ethics board. Consent was given by all patients. Surgeries were performed at the Montreal Neurological Hospital. Pre-operative magnetic resonance imaging was performed for surgical planning. Tumor samples were obtained at the junction of the contrast-enhancing portion of the tumor and brain invasion. In our experience, this location maximizes cell viability, reduces the confounding effects of hypoxia and necrosis, and increases the number of cells, which can be extracted from the sample. A certified neuropathologist confirmed all tumor histopathological diagnoses and IDH mutation status by DNA sequencing.

Cells were dissociated from the whole tumor, and cDNA libraries were prepared on the operative day (Supplementary Fig. 1a). Whole-tumor specimens were washed three times in sterile phosphate-buffered saline (PBS) containing penicillin and streptomycin. Specimens were then minced into fragments of <1 mm in size, before being digested in a collagenase solution containing DNAse and $MgCl_2$ for 1–2 h at 37 °C. The digested specimens were washed three times with sterile PBS, and large debris were removed with a 70-μm strainer. Residual RBCs were removed using a density gradient in a 1:1 volume ratio with the sample (Lymphoprep, Axis-Shield). Samples were washed five more times in sterile PBS.

**Preparation of the whole tumor and GSC samples**. The isolated cells were divided into two parts: one for whole-tumor analysis; and one for glioma stem cell enrichment.

Whole-tumor cells were prepared for single-cell capture and sequencing. For the early samples (Table 1), endothelial and myeloid cells were removed before capture. Later samples (Table 1) were captured and sequenced immediately after dissociation since normal cells were removed in silico. The isolated cells were resuspended at a concentration of 1e6/mL in PBS. After removing 50 μL as unstained control, the live/dead dye, Aqua (Molecular Probes) was added at a concentration of 1:1000. Cells were incubated for 25 min on ice, protected from light. Cells were washed once with PBS and resuspended in 100 μL of PBS with 1% bovine serum albumin (BSA). FcR block (Miltenyi) was added and incubated for 15 min. CD31 conjugated to BV421 (Biolegend), and CD45 (Biolegend) conjugated to PE were added to the suspension at pre-titrated values and mixed well by resuspension and incubated for 25 min on ice, protected from light before washing twice with PBS. Compensation beads (Molecular Probes) were used to prepare compensation controls for all antibodies and live/dead used. The sample was then resuspended in PBS with 5% BSA with 25 mM HEPES and 2 mM EDTA at a final volume of 300–500 μL and sorted on the FACS Aria III. Sorted cells were collected in polypropylene tubes with 1 mL of ice-cold FACS buffer with a temperature maintained at 4 °C throughout sorting. We selected cells that were negative for CD31 and CD45 (Supplementary Fig. 7a). Cells were resuspended in PBS with 0.04% BSA for single-cell capture (Supplementary Fig. 1a).

For GSC-enriched samples, whole-tumor presorted cells were expanded as neurospheres in complete neurocult-proliferation media (Neurocult basal medium containing: neurocult NS-A proliferation supplement at a concentration of 1/10 dilution, 20 ng/ml recombinant epidermal growth factor, 20 ng/ml, recombinant basic fibroblast growth factor, and 2 μg/ml Heparin) from Stem Cells Technologies. After 7 days of NCC culture, the neurospheres were collected in a tube and spun at 1200 rpm for 3 min. To dissociate the spheres, Accumax (Millipore) was added to the cell pellet and incubated for 5 min at 37 °C, they were then washed with PBS, centrifuged and resuspended in PBS with 0.04% BSA for single-cell capture (Supplementary Fig. 1a). GSC lines were proven to be tumorigenic by xenotransplantation.

**Human fetal brains**. Human fetal brain tissue samples (13–21 gestational weeks) were obtained from the University of Washington Birth Defects Research Laboratory (Seattle, Washington, USA), Centre Hospitalier Universitaire Sainte-Justine (Montreal, Quebec, Canada), and from the University of Calgary (Calgary, Alberta, Canada). These tissues were obtained at legal abortions. The use of the specimens following parental consent was approved by The Conjoint Health Research Ethics Board at the University of Calgary and studies were carried out with guidelines approved by McGill University and the Canadian Institutes for Health Research (CIHR). Cells were freshly isolated[60]. In brief, fetal brain tissue was minced and treated with DNase (Roche, Nutley, NL) and trypsin (Invitrogen, Carlsbad, CA, USA) before being passed through a nylon mesh. The flow cells were collected in PBS for sorting followed by sequencing (see below).

**Human adult brain**. Human autopsy brain specimens were obtained from de-identified excess diagnostic brain tissue that had been slated for incineration. Brains were cut in the coronal plane and immersed in 3% paraformaldehyde (PFA) or formalin for 1–2 weeks and then portions of their lateral ventricular walls were excised. These were further processed for immunolabeling, embedded in paraffin, and 5 μm thick sections were cut using a microtome (SLEE).

**Fetal cell sorting**. Dissociated fetal cells were washed thrice with excess ice-cold PBS and spun down at 1400 rpm for 10 min. Cells were resuspended at 1e6/mL of PBS and aqua live/dead dye (Molecular Probes) was added at 1:1000 and incubated for 25 min on ice, protected from light. Cells were washed once in excess PBS and were resuspended at 1e6/40μL and FcR block (Miltenyi) was added at 5 μL per 50 μL. Cells were mixed well and left to incubate on ice for 15 min. CD133-PE (eBioscience), CD45-PerCP/Cy5.5, and CD31-PerCP/Cy5.5 were added at a concentration of 1:20 and cells were resuspended well before being left to incubate on

ice for 25 min. A total of 1e5 cells were kept aside as unstained control and 5e5 cells were kept aside for fluorescence minus-one gating for CD133 only (FMO-PE).

All cells were washed twice with excess PBS and were spun down at 1400 rpm for 5–10 min. Cells were resuspended in ice-cold FACS buffer (5% BSA in PBS with 1% penicillin–streptomycin) before sorting. Sorted cells were collected into polypropylene tubes with 1 mL of ice-cold FACS buffer with a temperature maintained at 4 °C throughout sorting. All samples were acquired on the BD FACS Aria Fusion III.

Compensation beads (Invitrogen) was used to prepare compensation controls for all antibodies and live/dead stains used. A minimum of 5000 events were acquired for compensation matrix calculation and a minimum of 50e4 total events were collected for fetal samples and analyzed using FlowJo(v10, FlowJo LLC).

**Single-cell RNA sequencing.** For each sample, fetal or cancer, an aliquot of cells was taken and stained for viability with calcein-AM and ethidium-homodimer1 (P/N L3224 Thermo Fisher Scientific).

Following the Single Cell 3' Reagent Kits v2 User Guide (CG0052 10x Genomics)[34], a single-cell RNA library was generated using the GemCode Single-Cell Instrument (10x Genomics, Pleasanton, CA, USA) and Single Cell 3' Library & Gel Bead Kit v2 and Chip Kit (P/N 120236 P/N 120237 10x Genomics). The sequencing ready library was purified with SPRIselect, quality controlled for sized distribution and yield (LabChip GX Perkin Elmer), and quantified using qPCR (KAPA Biosystems Library Quantification Kit for Illumina platforms P/N KK4824). Finally, the sequencing was done using Illumina HiSeq4000 or HiSeq2500 instrument (Illumina) using the following parameter: 26 bp Read1, 8 bp I7 Index, 0 bp I5 Index, and 98 bp Read2.

Cell barcodes and UMI (unique molecular identifiers) barcodes were demultiplexed and single-end reads aligned to the reference genome, GRCh38, using the CellRanger pipeline (10X Genomics). The resulting cell-gene matrix contains UMI counts by gene and by cell barcode.

**Analysis of CNAs and identification of non-cancerous cells.** Cells from all samples were pooled in silico. The raw counts of each cell were first normalized using a trimmed mean of M-values (TMM) normalization approach[61]. This normalization is not affected by outliers but ensures that the majority of the genes support the normalization scale factor. The genome was then tiled by merging consecutive genes into expressed regions with a minimum average expression across the cells (five reads). This new expression matrix was also TMM normalized. For each region and each cell, a Z score was then computed by subtracting the average expression across cells and dividing by the standard deviation. These Z scores were winsorized at −3 and 3, minimizing the effect of strong outliers. To focus on the effect of CNAs, we minimized expression patterns that are specific to a single expressed region by applying a moving median. Using a sliding window of seven regions, this moving median approach replaced the expression of a region by the median over the surrounding seven regions (three upstream and three downstream).

A PCA was performed on the smoothed Z scores using non-cycling cells (see "Cell cycle and principal components analysis"). Because of the genome tiling and moving median, this PCA focuses on expression variability affecting large regions, hence driven by CNA. Louvain clustering was then performed on the K-nearest neighbour graph built using $K = 100$. The similarity between nodes was computed as $1/(1 + D)$ with $D$ the Euclidean distance on the first 20 principal components. "KNN" and a modified version of "igraph" R packages were used, respectively, for the KNN graph and Louvain clustering[62,63]. We scanned the resolution parameters $\gamma = 0.1$ to $\gamma = 1.5$, in increments of 0.1. We ran the Louvain clustering 100 times for each resolution, shuffling the order of the nodes in the graph each time. To assess the stability of the clustering at each resolution, we computed the average and standard deviation of the Adjusted Rand Index between pairs of classification[64] (Supplementary Fig. 1c). $\gamma = 0.2$ was the resolution with the highest average Rand index and lowest standard deviation[65]. T-distributed stochastic neighbour embedding[66], or tSNE, was used to visualize the cells across patients and clusters, using again the first 20 principal components.

Cells were annotated as normal if belonging to one of two communities each containing cells from almost all patients (Fig. 1a and arrows in Supplementary Fig. 1d). These clusters had low cycling scores and could not be explained by differences in sequencing depth. In addition, these two clusters formed an outgroup when focusing on chromosomes 7 and 10, two chromosomes that are known to host recurrent CNAs in glioblastoma[19]. As expected, normal cells had lower expression in chromosome 7 and higher expression in chromosome 10. The two clusters of normal cells were remarkable for their expression of myeloid genes in one cluster, and oligodendrocyte and endothelial genes in the other (Supplementary Fig. 1e), which indicates their nature. The presence of two clusters of normal cells is most likely due to subtle cell type specific patterns that were not fully corrected by the moving median (see example in Supplementary Fig. 7d).

Clones within tumors were defined by running the same Louvain clustering approach separately on the tumor cells of each patient. Here, the number of principal components used were automatically chosen by the "quick.elbow" function of the "bigpca" R package. The optimal resolution gamma was chosen as described above (Supplementary Fig. 1g). When the best average Adjusted Rand Index was lower than 0.7, we considered the clustering too unstable and grouped all

the cells from the patient into one unique clone. To characterize the CNA profile of each clone, cells were merged into supercells by summing their raw gene counts. For each clone, we created 10 supercells, each by merging 30 randomly selected cells. Supercells from normal cells were created similarly and were used later as baseline. The supercells for each clone were then pooled, TMM normalized genes were merged as above to create expressed regions with at least 20 reads on average. For each expressed region and each supercell, a log-ratio was computed by dividing the normalized counts by the average counts in the normal supercells. Using the log-ratios and a multivariate Gaussian mixture hidden Markov model (HMM), regions were classified as loss, neutral, or gain. The HMM had three states with means log(0.5), 0 and log(1.5), the empirical standard deviation estimated from the data, and represented the 10 supercells simultaneously for each clone. The "viterbi" function from "RcppHMM" R package was used to estimate the most likely states of a GHMM object. The transition probability was set to $10^{-40}$. We define a loss (gain) of a chromosome if more than 50% of the regions are in the loss (gain) state. Finally, the significance of each chromosomal CNA was confirmed using a Wilcoxon test on the median chromosome expressions. All CNAs showed $p$ values below 0.001. The HMM analysis was also run on the normal cells from each patient with the normal cells from other patients as baseline; no CNAs were detected (Supplementary Fig. 1h). We used a Chi-squared test to compare the cell distribution across the four TCGA signatures between pairs of clones (Supplementary Fig. 2e). Except for the first clone of BT333, clones had significantly different TCGA signatures ($p < 0.01$).

**Signal processing for transcriptional data.** Low complexity cells (<1000 genes or <1800 UMI detected), dying cells (>12% UMI to mitochondrial genes, Supplementary Fig. 7e), non-cancerous cells (see Analysis of CNAs and isolation of non-cancerous cells) and genes with no counts were removed from the analysis. Next counts were adjusted in each cell according to a size factor akin to TPM. Genes that accounted for >1% of UMI in a given cell were not counted towards the UMI sum of this cell. Similar to previous studies[32,33], each cell was normalized to 1e5 UMI.

Signal-containing, non-random genes were selected in each sample. This was done in a manner similar to that described by Klein et al.[32]. In brief, we selected the 3000 highly variable genes in every sample by Fano statistic. We then applied a base 2 logarithm to obtain the normalized expression matrix. A z score by gene was applied at this point for single sample analyses. For analyses spanning multiple samples, we combined the normalized expression matrices on the basis of the intersection of their significant genes. Z-score across all cells and samples was applied by gene thereafter.

**Filtering the fetal brain and cancer samples.** We removed ependymal cells and microglia from later fetal analyses. These were seen as separate clusters in PC1 and PC2 in most samples. Microglia had high expression of genes such as P2RY12 and CX3CR1[67,68], whereas ependymal cells had high expression of SPAG6, FOLR1, and FOXJ1[69,70].

BT346 contained many cells with a signature not seen in other samples. These clustered separately in tSNE and PCA. We used k-means ($k = 2$) to separate them from the other cells. A GSEA (see "Quantification and Statistical Analysis for methodology") showed that the top four most significant gene sets were linked to hypoxia (e.g., HALLMARK_HYPOXIA, MENSE_HYPOXIA_UP). This tumor was unique in that the magnetic resonance imaging region of contrast enhancement was very thin. It is thus likely that some cells from the necrotic core were isolated. We excluded the hypoxic cells in BT346 from later analyses and did not include them in the total cancer cell number reported.

**Cell cycle and principal components analysis.** We positioned all cells within the cell cycle according to the method presented by Tirosch et al.[24]. In brief, each cell obtained a score for the G1/S phases and a score for the G2/M phases (Supplementary Fig. 7f). A list of genes deemed characteristic of those cell cycle states was used[24]. Each score was defined as the sum of the expression of all genes within its corresponding gene set, then z scored across cells. As most cells are not cycling (Supplementary Fig. 7f), we defined non-cycling cells as those with both G1/S and G2/M scores <0.

Cycling-free PCA was performed for each sample individually as follows. The PCs, or eigenvectors of the covariance matrix, were obtained from the non-cycling cells only (as defined above). We then use these cell cycle-independent eigenvectors to project the complete data set in PCA space (Supplementary Fig. 2a).

The first PC of every GSC sample was highly conserved (see Results). To quantify this, we compared the ranking of genes by PC1 loadings across samples. The actual ranking of each gene was obtained in all samples. To obtain the mean ranking, the actual rankings were averaged by gene, and these averaged values were then ranked. For each gene, we thus obtained five actual rankings (one per sample) and one mean ranking. $R^2$ was obtained by least-square linear regression in PC1 and PC2 separately (Matlab, *fitlm*).

**Classifying cells by TCGA subtype.** TCGA subtype for each whole-tumor cell was obtained by scoring each cell for their proximity to each TCGA centroid[19]. The highest score obtained by a given cell defined the subtype of the cell. We used this

method on the original TCGA data set and found we could correctly classify 89.7% of all tumors.

Proximity is calculated as follows. The position of a cell in the TCGA transcriptomic space ($X_{ij}$) is obtained from the expression of the genes present in the TCGA signature ($S$). The unit vector of this cell's position is then projected onto the unit vector of the signature of interest using a dot product.

$$P_i = \frac{X_i \cdot S}{|X_i||S|}$$

where $P$ is the projection score and $S$ is the signature of interest.

We also subtyped cells using the more recent TCGA signatures[20] and classifier[71]. We randomly selected 1000 cells from our data set and entered their data for all non-zero genes in the Gliovis data portal.

**Clustered non-negative matrix factorization.** The cNMF algorithm was applied individually to each whole-tumor sample[35], with some modifications. In brief, non-negative matrix factorization was run (nmf, Matlab, multiplicity algorithm, replicates = 20, maximum iterations = 1e6) 100 times for $k$ from 2 to 15 signatures. For each $k$, the 100 repetitions are clustered in k groups. We expect a stable clustering solution would produce tight clusters with one signature per cluster for each of the 100 repetitions. The proportion of repetitions with one signature per cluster was called reproducibility. Clustering of the signatures was done by $k$ means (Matlab, using correlation) with a constraint of uniform cluster sizes, prioritizing higher correlations. The largest $k$ with a reproducibility above 0.9 was chosen (Supplementary Fig. 7g, left plot). For a chosen $k$, we confirmed the clustering solution was appropriate by running tSNE on the signatures it generated (Supplementary Fig. 7g, right plot). The final signatures for a given sample was obtained by averaging the signature repetitions within a cluster, excluding repetitions with poor reproducibility (the ones which did not produce a signature per cluster).

We obtained between five and nine final signatures per sample, 79 signatures in total. From the inter-signature Pearson correlations, we used hierarchical clustering to find trends of signatures (Fig. 1e, hierarchical tree). Six main groups emerged. In one of these groups, important variations in gene weights were observed: signatures characterized by OLIG2 and ASCL1, for example, had less DCX and STMN1, and vice versa. We reclustered this group in two, yielding the final seven groups (Fig. 1e and Supplementary Fig. 2f).

We identified the most characteristic genes of each signature group by ranking genes according to their relative signal to noise ratio (snr) and chose the top 40 for the heatmap.

$$snr = \frac{\mu_{group} - \mu_{not}}{\sqrt{\sigma_{group}^2 + \sigma_{not}^2}}$$

We scored each signature according to the TCGA using the method described above (Classifying cells by TCGA subtype). A given signature was labeled with the subtype yielding the highest score (Fig. 1e).

**Community detection in fetal samples.** To properly cluster fetal cells in cell types, the modular structure of the gene coexpression network was estimated using community detection. Data from all fetal samples were merged as explained above. PCA was performed on the merged data set (see Principal components analysis above). The first 10 PCs were selected based on the importance of their corresponding eigenvalue (Supplementary Fig. 7h). The connection weights were computed as $1/(1 + D)$ with $D$ as the Euclidean KNN graph between nodes in this PC space, with $K = 50$. Self-weights were set to 0 to promote the formation of communities.

Again, the goal of the analysis was to identify groups of cells that are more similar to each other than other cells. This constraint was operationalized in terms of the modularity $Q$[62]:

$$Q(\gamma) = \sum_{ij} \left[w_{ij} - \gamma p_{ij}\right] \delta(c_i, c_j)$$

where $w_{ij}$ is observed connection weight between nodes $i$ and $j$, whereas $p_{ij}$ is the expected connection weight between those nodes. The Kronecker delta function, $\delta(c_i, c_j)$ is equal to 1 when nodes $i$ and $j$ and assigned to the same community ($c_i = c_j$) and zero otherwise ($c_i \neq c_j$), ensuring that modularity is only computed for pairs of nodes belonging to the same community. The resolution parameter $\gamma$ scales the importance of null model $p_{ij}$, potentiating the discovery of larger ($\gamma < 1$) or smaller communities ($\gamma > 1$)[63].

In the present study, the expected connection weight between pairs of nodes was defined according to a standard configuration model, such that:

$$p_{ij} = s_i s_j / 2m$$

where $s_i = \sum_i w_i$ is the strength of node $i$ and $m = \sum_{i,j>1} w_{ij}$ is total weight of connections. Under this null model, communities are considered to be of high quality if the constituent nodes are more highly correlated with each other than in a randomly rewired network with the same strength distribution and density.

The quality function $Q$ was maximized using a Louvain-like locally greedy algorithm[72], as implemented in the Brain Connectivity Toolbox

(community_louvain.m)[73]. We scanned the resolution parameters $\gamma = 0$ to $\gamma = 1.5$, in increments of 0.1. At each scale, the Louvain algorithm was run 100 times to find a partition that maximized the modularity function[72].

To select an appropriate scale, we computed the $z$ score of the Rand index between all pairs of partitions at each scale[64]. We selected the resolution at which the mean pairwise Rand index to standard deviation ratio (SNR) was greatest across the partition ensemble[65]. The logic behind this approach is that if there exists a particularly well-defined community structure at some topological scale, then it should be relatively easy to detect, and the partitions will not vary greatly across runs. Supplementary Fig. 3b shows the SNR of all pairwise Rand indices. Based on this method, we selected $\gamma = 1.0$. Once the scale was selected, we used the consensus heuristic described by Bassett et al.[65] to find the most representative partition in the ensemble (Brain Connectivity Toolbox; consensus_und.m).

Two modifications were made to this solution. The first was to consolidate excitatory neurons–four clusters coincided on the tSNE plot and strongly expressed neuronal genes such as NEUROD6, SYT1, and STMN2. Second, a smaller cluster spanned multiple apparent groups on the tSNE plot. Differing expression of OLIG2, PDGFRA, APOD, GFAP, SOX9, APOE, ASCL1, and MKI67, among other genes, were apparent within this cluster (Fig. 2c dotted circle, and Supplementary Data 1). The algorithm described above was used again on this group of cells, with resolution parameters scanned from $\gamma = 0$ to $\gamma = 1.5$, in increments of 0.01. Lower increments were used because less total nodes allowed for additional computational time. $\gamma = 0.44$ was the consensus or most representative partition. This further separated it into three clusters consistent with: OLCs; astrocytes; and a previously unidentified glial cell type we called glial progenitor cells.

The values shown in the similarity matrix heatmap (Fig. 2b) are the inverse of the diffusion pseudotime[41] between cells.

**Differential expression of fetal cell types and comparison with a reference fetal data set.** We assessed differentially expressed genes between fetal brain cell types by comparing each cell type with all others combined. A Mann–Whitney $U$ test (Matlab, ranksum.m) was applied on the log expression value (before $z$ score) of each gene sequentially. $P$ values were adjusted for multiple testing using the approach described by Storey[74], and are reported as $q$ value (Matlab, mafdr.m).

To compare our data set to the reference data[38], we used two complementary approaches: cluster-based comparison and a cell-based scoring approach.

The cluster-based approach uses signatures (markers) of each cluster in the two data sets. A similarity (distance) matrix was computed with the Jaccard Coefficients (JC, fraction of shared markers) using a maximum of 100 most significant genes for each cluster in both studies (with $p$ value <0.01). Using the JC matrix, with reference clusters in rows and current clusters in the columns, a heatmap was generated (Supplementary Fig. 3c) using pheatmap, R package, with default parameters and no re-ordering of the columns (cluster_col=FALSE).

For the cell-based approach we scored each cell for sets of markers from clusters in the reference study[38]. The AUCell[75] ranked-based scoring method was used, which is independent of the gene expression units, in order to score cells using markers ($n$ marker = 60) for each reference cluster. The scores were then visualized across all clusters. The nine reference clusters (out of 47) that help most to annotate current clusters have been shown in (Supplementary Fig. 3d).

**Creation of the fetal roadmap.** We aimed to create a fetal roadmap, or a transformation of transcriptomic space descriptive of the transitions that exist between the cell types present in glioblastoma. Simply, we first determined which fetal cell types were most representative of the cancer; then we created a PC space of these cell types onto which the cancer could be mapped.

We determined the most representative cell types by finding each cancer cells closest transcriptomic fetal brain cell neighbour. Data from all whole tumor and fetal brain samples were merged as described above. Each cell type was randomly subsampled (Matlab, randsample.m) to obtain an equal number of cells for each cell type. Whole-tumor samples were similarly subsampled. The top 10 PCs for this new fetal data set were calculated (see "Removal of cell cycle" section), and both fetal brain and cancer data sets were projected in this space. The closest fetal brain cell neighbour for each cancer cell was found (Matlab, knnsearch.m). We refer to this as a capture of this cancer cell by the fetal cell. The number of cancer cells captured by each fetal cell type was tabulated. Neuronal progenitor cell types were tabulated under their more-differentiated counterpart.

The proportion of enriched GSCs captured by GPCs was substantially greater than that of whole-tumor cells (46% vs 24%, Supplementary Fig 4a, b), whereas only 0.6% of enriched GSCs were captured by tRG compared with 11.1% in whole-tumor cells (Supplementary Fig. 4b). Some neuronal cell types also captured cancer cells, as predicted by the cNMF (Fig. 1e) and GSC (Fig. 1b) analyses.

Five fetal brain cell types were retained for the creation of the cancer roadmap. As was done above, fetal brain cells from these five subtypes were randomly subsampled to balance their numbers. Genes common to both cancer cells and fetal brain cells were kept for the analysis ($n = 398$ for whole tumor, $n = 401$ for GSC, and $n = 345$ for GSC and whole tumor combined) and $z$ score normalization of the log counts was done on the complete data set. Cell cycle-free PCA was performed on these fetal brain cells (see above) and 10 PCs were kept. We defined this as the roadmap. To refine this separation and better capture the transitional nature of this data, we performed diffusion embedding on the roadmap. In brief, from the

roadmap we calculated a transition matrix[41] (*diffusionmap.T_nn.m*, $k = 50$, *nsig* = 10). The top five eigenvectors were obtained and normalized (*eig_decompose_normalized.m*). The first eigenvector was dropped as the steady state of the transition matrix. Eigenvectors 2 to 5 were studied for their ability to resolve all cell types. Eigenvectors 2, 3, and 5 were defined as DC1, DC2, and DC3, respectively. The glial progenitor score was defined as DC3.

**Mapping of cancer cells to the fetal roadmap**. The aim of the roadmap was to highlight the underlying hierarchical organization while de-emphasizing inter-patient variability. Hence, we projected cancer cells (whole tumor, GSCs, or both) onto the 10-dimensional fetal PC space of the roadmap. This represents the mapping of cancer cells onto fetal PC space. We used these results to obtain the diffusion and the simplified PC mapping of cancer, as we will explain below.

To obtain cancer mapping in diffusion space, we first obtained the transition matrix of the fetal roadmap as described above. From the PC cancer mapping, a separate transition matrix was obtained for cancer but solely as a function of the fetal brain cells (*diffusionmap.T_nn.m*, k = 50, nsig = 10). The cancer transition matrix ($T_{cancer}$) was then projected onto the roadmap DCs ($\phi_{fetal}$) defined above.

$$\phi_{cancer} = T_{cancer}\, \phi_{fetal}$$

The resulting DC vectors ($\phi_{cancer}$) represented the mapping of cancer cells in the roadmap diffusion space.

To rule out the possibility that this hierarchical distribution could be the product of chance, we created control cells by randomly swapping the genes in our whole-tumor cells. These control cells would have had the same depth of sequencing, but gene signatures were absent. Using a Kolmogorov–Smirnov statistic, we found that our cancer cells and control cells had a very significantly different distribution when projected on the roadmap, both in diffusion space and PC space ($p$ value < 1e-22 for both).

Next, we sought to create a simplified PC roadmap, in an effort to better capture biological relevance. This is because both GPC and OLC populations contain progenitors, and a reliable surface marker to differentiate the two was not found. In the PC roadmap, PC2 and PC4 separate fetal OPC and fetal GPC from the other cell types, respectively. Therefore, to make a combined progenitor score, we summed the values of PC2 and PC4 (Fig. 4c). PC1 already separated astrocytes/tRGs (positive values) from interneurons (negative values). We defined the latter as our lineage score. Cancer genes which correlated with each of these two scores (see Supplementary Data 3) guided our search for markers for each cell type.

**Classification of cancer cells by cell type**. In order to compare the signatures of cells at the extreme ends of the hierarchy, we aimed to classify the cancer cells by cell type. Using the annotated fetal data in diffusion roadmap space as a training set, we performed a LDA (Matlab, *fitcdiscr.m*). Cancer cells in diffusion roadmap space were classified using this model (Fig. 3b). In order to classify extremes of the hierarchy only, any cell with a probability of incorrect classification of more than 0.01% was left unclassified.

**Similarities between cell types and signatures**. Similar to the comparison with a reference fetal data set described above, we first scored each cell for the sets of markers (signatures) to compare. For example, when comparing Neftel et al.[42] with our roadmap results, we used the cell states from Neftel et al.[75] and scored all cells in our data set as explained above. Then for each cell, the best ranked score, across all signatures, was consider as the predicted cell type. In the next step, these predicted cell types were compared with our original roadmap cell types and reported and visualized as proportion summing up to 1 (Supplementary Fig. 4f). The method was also applied for the comparison of the roadmap to the cNMF signatures (Supplementary Fig. 4f). Finally, the same approach was applied to compare our signatures from our roadmap and Neftel et al.[42] to the TCGA data set (Supplementary Fig. 4g). We obtained similar results to what was reported by the authors. Roadmap markers for each cell type were obtained using FindAllMarkers function in Seurat package version 2.3.4 with default parameters and p value < 0.01[76].

**RNA velocity of cancer cells**. We performed this analysis as described by La Manno et al.[46] using the Velocyto package on whole-tumor samples with one thousand cells or more (10 samples) and analyzed them independently using modifications described here. We calculated spliced and unspliced counts using the Velocyto package as described, merged all patient data sets, and removed normal cells based on the results of our previous analysis (see Identification of non-cancerous cells above). Gene selection was performed with Velocyto in a manner similar to our previous analyses: minimum expressed counts of 40 in a minimum of 30 cells for the top 3000 most variable genes. Spliced and unspliced counts were normalized separately, but no count imputation was used. Knn imputation had the unwanted effect of amplifying patient-specific properties as cancer samples did not overlap in transcriptomic space. We then calculated a gene-wise mutual information between spliced and unspliced counts. We excluded genes below the 10th percentile of mutual information. This did not change the trends observed in the results but did remove some noise. The gene-specific steady-state constants (gamma) were then calculated from the resulting data set with all patients

combined. Similar results were obtained when gamma was calculated separately for each patient (data not shown). RNA velocity and extrapolated cell states (with $t = 1$) were then estimated as described. Thereafter, the data set was separated by sample. Embedding was obtained using Uniform Manifold Approximation and Projection (UMAP, *sklearn.feature_selection*)[77] or using PCA. UMAP was done with correlation as the metric, no prior dimensionality reduction, minimum distance set to 0.4, and number of neighbors set to one tenth the number of cells in the sample. PCA was performed after the normalization done by the Velocyto package. Transition probabilities and embedding shifts were measured using all cancer cells in a given sample. We ensured the results were not the product of random chance by comparing them to that of randomized data (Velocyto, Supplementary Fig. 5b for a representative example). Arrows were plotted on an absolute scale.

The embedding scatter and velocity quiver plots were overlaid, and the colors were given based on the cell types attributed in the fetal roadmap analysis (see Classification of cancer cells by cell type), with the exception of progenitor cells and unclassified cells. These cells were instead colored according to a greyscale proportional to their position in the glial progenitor axis (Fig. 5a). This was done to better visualize the continuity of differentiation in the embedding.

**Pathway enrichment for progenitors in whole tumor**. Whole-tumor cells classifications were obtained using the LDA method described above. Progenitor and astrocytic/mesenchymal classifications were used. As had been done previously[48], each gene was ordered according to its signal to noise ratio (SNR) for the progenitor vs the astro-mesenchymal cell types

$$SNR_j = \frac{\hat{\mu}_{jP} - \hat{\mu}_{jA}}{\hat{\sigma}_{jP} + \hat{\sigma}_{jA}}$$

where $\hat{\mu}_{jX}$ is the estimated mean log expression of gene $j$ for progenitor ($P$) and astrocytic ($A$) cancer cells; and $\hat{\sigma}_{jX}$ is the estimated standard deviation of log expression for gene $j$. A Mann–Whitney $U$ test (Python, *scipy.stats.mannwhitneyu*) was used to determine if the $SNR$ values for genes in a given gene set were significantly different than the $SNR$ not in this gene set (Supplementary Fig. 7i for an example). All gene sets in the *c2.all.v6.0* data set from the Broad Institute[48,78] were tested, using the genes present in our combined whole-tumor data set ($n = 970$, see Supplementary Data 2 for list of genes). $P$ values were adjusted for multiple testing using the approach described by Storey[74], and reported as $q$ value.

**Mass cytometry**. Metal tagged mass cytometry antibodies were purchased from Fluidigm. Where tagged antibodies were not available, purified antibodies lacking carrier proteins were labeled with heavy metal loaded maleimide conjugated DN3 MAXPAR chelating polymers (Fluidigm) according to the recommendations provided by Fluidigm.

Cells were stained according to a well-established protocol for cell cycle staining[44]. In brief, cells were incubated with IdU at 50 μM final concentration for 30 min at 37 °C and 5% $CO_2$ in stem cell media. A live/dead stain was performed by incubating cells with 5 μM cisplatin (Fluidigm) at room temperature for 5 min. Cells were washed twice with cell staining buffer (CSB), composed of standard PBS with 0.5% BSA and 0.02% sodium azide, twice. Before cell surface antibody labeling, Fc-receptors were blocked using human BD Fc block (BD biosciences). Cells were then labeled with a surface antibody panel which included CD9, CD24, CD44, CD133, PDGFRα, HLA-ABC, Olig2 and CD45, and CD31 and incubated on ice for 25 min. Cells were then washed and fixed using Fix I buffer (Fluidigm) for 15 min. This was followed by two more washes with CSB and ice-cold methanol fixation for 15 min on ice. Intracellular labeling was carried out for 25 min on ice. A final two more washes with CSB were carried out followed by an overnight incubation in Fix and Perm buffer (Fluidigm) with 125 nM of iridium intercalating dye (Fluidigm).

Mass cytometry data were analyzed using FlowJo (v.10, FlowJo LLC) and a hyperbolic arcsine transformation on all parameters after filtering out dead cells and CD45 or CD31-positive cells.

**Glioma stem cell sorting**. Multiparametric flow cytometry was carried out by labeling cells with CD9 preconjugated with BV421 (BD Pharmingen), CD24 preconjugated with APC or APC-H7 (Miltenyi), CD44 preconjugated with AF700 (BD Pharmingen), and CD133/PROM1 preconjugated with PE or PE/Vio770 (eBioscience and Miltenyi). After leaving aside 1e5 cells as unstained control, cells were resuspended in PBS at a concentration of 1e6/mL. Aqua live/dead dye (Molecular Probes) was added at 1:1000 and incubated for 25 min on ice, protected from light. Cells were washed and 1e5 cells were kept aside for fluorescence minus-one (FMO) controls and 1e6 cells were used for complete staining with antibodies. FMO controls were prepared for all colors except aqua (live/dead). All cells were completely stained with antibodies at a final dilution of 1:50-1:20. FMO controls were used to identify for positive/negative staining.

Sample preparation post-staining for sorting and data acquisition was carried out as described above. Gating strategies are shown in Supplementary Fig. 7b, c.

**Luciferase vector**. The Red Firefly Luciferase sequence was amplified from the pCMV-RedFLuc (Targeting Systems, CA, USA) and cloned into the bidirectional

EF1/PGK promoter lentiviral vector (System Biosciences, Palo Alto, USA). The final construct was named PGK-GFP-LUC. Lentivirus was produced as per the protocol described by Ritter et al.[79]. Expression of the construct was validated by luciferase assay and fluorescence microscope.

**Mouse xenotransplantation**. All animal procedures were approved by the Institution's Animal Care Committee and performed according to the guidelines of the Canadian Council of Animal Care. We orthotopically injected 100k (for general tumorigenicity and E2F inhibition) or 5k (cluster tumorigenicity) GFP[+]/Luciferin[+] GSCs into female NOD-SCID gamma mice (Charles River, Wilmington, USA)[80,81]. In brief, mice were anesthetized at 5 weeks of age using isofluorane (Fresenius Kabi, Bad Homburg, Germany) and placed on a stereotaxic apparatus. A midline scalp incision was made and a burr-hole (3 mm) was created 2.2 mm lateral to the bregma using a high-powered drill. The injection needle of a Hamilton syringe (Hamilton, Reno, USA) was then lowered into the burr-hole to a depth of 2.5 mm and cells were transplanted into the striatum. Animals were frequently monitored and then killed at the appearance of distress signs and/or 10% weight decrease. These animals were perfused with PBS and their brain collected. Kaplan–Meier curves were created according to the survival results. A Cox proportional hazard ratio model was used to assess significance, with patient cell line and cell type (Fig. 6e) or treatment group (Fig. 7f) as covariates. This analysis was performed in R using the packages *splines* and *survival*. For the treatment experiment (Fig. 7f), cells were treated in vitro with drug or vehicle, and 100k live cells were injected on the third day of treatment for each treatment group.

Harvested brains were placed in 10% neutral buffered formalin for 72 h at room temperature. After formalin fixation, specimens were processed and paraffin-embedded. Five µm tissue sections were prepared and mounted on a poly-L-lysine-coated glass slides for subsequent analysis.

**In vivo imaging**. To monitor tumor growth, we imaged each mouse every 2 weeks using the In Vivo Imaging System Sprectrum (Perkin Elmer, Waltham, USA) according to the manufacturer's instructions. In brief, we intraperitoneally injected a solution (15 mg/ml) of luciferin (Perkin Elmer) at the dose of 150 mg/kg, and after 3 min, mice were anesthetized using isofluorane. At 10 min from luciferin injection, we positioned the mouse in the imaging system and began image acquisition. The exposure time was automatically determined by Living Image 4.5.2 software (Perkin Elmer). Results are reported as number of photons emitted, and a two-sample student's *t* test was performed, two-sided.

**Immunofluorescence**. GSCs were grown on laminin (10 µg/ml) coated coverslips, and fetal neural stem cells were grown on Matrigel in the supplemented mTeSR1 basal medium (STEMCELL Technologies). Both were fixed with 3% PFA and permeabilized with 0.5% Triton X-100 before being immunolabeled with indicated antibodies followed by secondary antibodies. Coverslips were mounted on glass slides using ProLon Diamond Antifade Mountant with DAPI (Invitrogen) to counterstain cell nuclei. Fluorescent images were acquired using ZEISS LSM 700 laser scanning confocal microscope with a ×20 or ×63 objective.

For the GSC assays, the total number of Ki67[+] cells relative to total cell number were quantified from 10 fields for each patient cell line (*n* = 3).

For tissue sections (brain and tumor) immunohistochemistry, samples were baked overnight in a standard laboratory oven at 60 °C, then deparaffinised and rehydrated using a graded series of xylene and ethanol, respectively. Antigen retrieval was done using citrate buffer (pH 6.0) for 10 or 20 min at 120 °C in a decloaking chamber (Biocare Medical). The slides were then blocked for 20 min with a commercial protein block (Spring Bioscience), incubated overnight at 4 °C with indicated antibodies, then slides were washed with IF buffer (PBS+0.05% tween20+0.2% triton X-100), following by incubation (1 h at room temperature) with according secondary antibodies (Invitrogen). Coverslips were mounted on glass slides using ProLong Diamond Antifade Mountant with DAPI (Invitrogen) to counterstain cell nuclei. Fluorescent images were acquired using ZEISS LSM 700 laser scanning confocal microscope with a ×63 objective.

For tumors, total number of CD133[+] or Ki67[+] or both CD133[+] and Ki67[+] cells relative to total cell number were quantified from at least 10 images from each patient. A $\chi^2$ test was performed to obtain the level of significance. A significant association of Ki67 and CD133 was found in all patients.

For xenografts, quantification was based on capturing 10–20 high-powered images per slide from multiple slides from each mouse per GSC type implanted from the 12-week cohort. For the apical GSC mice 2712 cells were counted, for the neuronal GSC mice 1261 cells were counted, and for the astro-mesenchymal GSC mice 2720 cells were counted. Error bars were measured as the standard error between HPF for each GSC cell type.

Primary antibodies used: anti-GFAP (Abcam); anti-Olig2 (EMD Millipore), anti-Ki-67 (Invitrogen and Abcam), and anti-CD133 (Miltenyi Biotec), anti-ASCL1 (Abcam), anti-CD44 (EMD Millipore), and anti-DCX (Abcam).

**Chemotherapy and targeted therapy assays**. TMZ–GSCs from each cluster type were seeded on laminin (10 µg/mL, Sigma) at a concentration of 10,000 cells/well in a 96-well plate and were subsequently treated for 5 days with varying concentrations of TMZ (Sigma Aldrich) ranging from 1 µM to 750 µM. In all, 50 µL of XTT

was prepared according to the manufacturer's instructions (Life Technologies), and the XTT mix solution was added to the cells and further incubated for 3 h at 37 °C. The absorbance at 450 nm was measured on an Epoch Microplate Spectrophotometer (Biotek Instruments, USA).

HLM006474–GSCs from each cluster type were plated on laminin (10 µg/mL, Sigma) at 5000 cells/well in 96-well plate overnight in culture media. The following day, HLM006474 (or DMSO) was added to a 10 µM final concentration in a final volume of 200 µL. Following 7 days of incubation at 37 °C, an XTT was performed as described above.

Combination therapies–GSCs (BT326-GSC) were plated on laminin (10 µg/mL, Sigma) at a concentration of 7000 cells/well in a 96-well plate and treated with either TMZ (50–450 µM) for 6 days, HLM (2.5–10 µM) for 6 days, or HLM006474 for 3 days followed by TMZ for 3 days. After these 6 days of treatment at 37 °C, an XTT assay was performed as described above. For the isobolographic analysis, 40% efficiency isoboles were found for all biological replicates (*n* = 3). A curved reference isobole was used because the maximum efficiency of HLM006474 is significantly higher than that of TMZ[82]. *P* value was calculated using a Student's *t* test.

Sphere forming assay–GSCs from each cluster type were plated at 150,000 cells/ well in six well plates with 20 µM HLM006474 in a final volume of 3 ml. After 7 days, cells were imaged with ×10 objective with Invitrogen EVOS FL/FL color microscope. Sphere diameter measurements were made with Image J. 6502 spheres were measured in two different patient GSC cell lines. An arbitrary cutoff for big and small spheres was set at 65 µm. A multivariate logistic regression was used to assess the likelihood of finding big spheres in each of the different GSC cell types treated, using patient cell line and cell type as variables. There was no significant difference between patient cell line (*p* = 0.69). An analysis of odds ratio is depicted as a forest plot in Fig. 7c.

All assays were performed in three different patient cell lines in three or more different cell passages and five technical replicates. *P* values describe differences in cell types and were calculated using a two-sample *t* test, two-sided. Stock solutions of TMZ (Sigma Aldrich), HLM006474 (Tocris-Bioscience) were prepared in dimethyl sulfoxide (DMSO; Sigma Aldrich), and were added to cells for a final DMSO concentration of <0.1%.

**Reporting summary**. Further information on research design is available in the Nature Research Reporting Summary linked to this article.

## Data availability
The single-cell sequencing data will be available on the European Genome-Phenome Archive: EGAS00001004422. Full western blots are provided as a source data file. All other data are available in the Article file, Supplementary Information or available from the authors upon reasonable request. Source data are provided with this paper.

## Code availability
All computations and quantifications were performed using Matlab, R, and Python programming languages. Scripts can be found at: https://github.com/mbourgey/scRNA_GBM. The single-cell CNV analysis package created in the course of this work can be found at https://github.com/jmonlong/scCNAutils. Source data are provided with this paper.

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

## Acknowledgements

We would like to thank Carmen Sabau for her contribution to the administrative work of the project, as well as Rozica Bolovan and Maryam Safisamghabadi for technical support. We are grateful to Sara Baig for the design of the schematic in Supplementary Fig. 1a, Greg Chang for supplying reagent for mass cytometry experiments, Jason Karamchandani for help accessing and formatting the TCGA data, Adrian Veres for advice with the initial phase of scRNAseq data processing, and Stefano Stifani for advice with interpretation of the fetal data sets. We also thank the McGill University and Genome Quebec Innovation Centre sequencing platforms, the bioinformatics platform at the Center for Computational Genomics, the Cell Vision Core Facility for Flow Cytometry and Single Cell Analysis, the Imaging Facility, the Histology Facility of the Life Science Complex of McGill University, the Flow Cytometry and Cell Sorting Facility of the Department of Microbiology and Immunology of McGill University, the Flow Cytometry Facility of the Institut de Recherches Cliniques De Montréal, the Flow Cytometry Facility of the C-BIGR of the Montreal Neurological Institute, the Mass Cytometry Core Facility of the Life Science Complex of McGill University, the Immunophenotyping platform of the RI-MUHC of McGill University. Funding provided by Compute Canada Resource Allocation Project wst-164 (to J.R.), CFI Leaders Opportunity Fund, Genome Canada Science Technology Innovation Centre, and Genome Innovation Node (all to G.B. and J.R.), the Cancer Research Society, Canadian Cancer Research Institute, Brain Tumor Foundation of Canada, and the Canadian Institute of Health Research (all to K.P.). We wish to thank the TARGiT Foundation, the A Brilliant Night Foundation, and the Argento Family Group Ercole for supporting this work. CPC is supported by the Fonds de Recherche du Québec–Santé Resident Physician Research Career Training Program Phase 1. SA is supported by a George H. Harris Fellowship and a MNI-Fisher Brain Tumor Award. G.B. is supported by a Fonds de Recherche du Québec–Santé Junior 2 Award. K.P. is supported by a clinician–scientist salary award from the Fonds de Recherche du Québec–Santé and the William Feindel Chair in Neuro-Oncology.

## Author contributions

C.P.C. and K.P. conceived the project, designed the study, and interpreted the results. K.P. performed the surgeries, and P.L. and X.Y. collected the single cells. Y.C.D.W. generated the single-cell sequencing libraries. C.P.C. performed scRNAseq-related computational analyses. J.M. performed CNA-related computational analyses. J.N. performed the fetal data set comparison to Nowakowsky et al., the signature comparison to Neftel et al., and the analysis of the TCGA data set. S.A. performed flow and mass cytometry. P.L., S.B., and C.L. performed immunofluorescence imaging. P.L, X.Y., R.A., and S.B. performed the drug and chemotherapy assays. R.A. performed the cloning of the luciferase vector and produced the lentivirus. X.Y., G.R., S.A., and C.P.C. performed the in vivo experiments. J.A. and V.W.Y. provided access to fetal brain samples. M.B., M.C.G., G.B., B.M., H.N., and J.R. provided analytical and experimental support. C.P.C., S.A., and K.P. wrote the manuscript with feedback from all other authors.

## Competing interests

The authors declare no competing interests.
