## [Peer review file · Nature Communications]

On receipt of Reviewer #2's report in the second round of review we sought an informal response from the authors and the informal comments of an additional expert on this response. The expert was of the opinion that the manuscript was suitable for publication in Nature Communications.

REVIEWERS' COMMENTS: Reviewer #2 (Remarks to the Author):

I commend the authors on their willingness to reevaluate their interpretation. In the revised analysis, the authors have chosen to include truncated radial glia, but not outer radial glia, or other radial glia cell types in their revised map. The authors continue to cherry-pick their analysis to support the hypothesis of an oligodendrocyte progenitor, and not a radial glia, as the glioblastoma cell of origin. This approach ignores much of their own data, as well as several recent important studies. I would ask the authors to consider the following points, which call into question the manuscript's technical correctness, in its current form:

1. The authors must compare the mesenchymal GBM signature to their developing brain data via a simple heatmap on an absolute scale, including the canonical mesenchymal GBM markers: CD44, CHI3L1, NAMPT, TNC, VIM. The "developmental roadmap" projection strategy is needlessly complex and hides the correlation between radial glia and the mesenchymal GBM signatures which a simple heatmap will reveal. I anticipate that the authors will find similar enrichments in their data as are found in the human fetal-brain data of Nowakowski et al. (below), which the authors used for mapping cell labels to their own data.

Nowakowski et al. via UCSC cell browser

2. The authors must include all radial glia cell types in their "development roadmap", including those labeled as RG and uRG by the authors, if they choose to retain their projection strategy at all. By only including tRG, the authors have specifically depleted genes co-expressed by both mesenchymal GBM stem cells and neural stem cells. This is clear from Fig. S4a-b, which shows that tRG are relatively depleted in GSC-enriched cultures while uRG are relatively enriched (yet excluded from the roadmap).

Beyond the cherry-picking, the projection approach is problematic overall, it is not clear why the cells were down-sampled, how the cells chosen were selected, and why that number of cells was used. I recommend the authors consider a simpler approach such as a heat map comparison between GBM cell-type signature genes (identified via simple PCA or clustering) and their fetal brain data.

3. In their revision the authors say:

Page 12, paragraph 3

Using this developmental data as a roadmap, we showed that IDHwt glioblastoma is hierarchically organized into three cell lineages that correspond to all three normal neural lineages: astrocytic; neuronal; and oligodendrocytic. Interestingly, the neuronal lineage is devoid of HLA expression suggesting a potential source of resistance to immunotherapy. We identified a fourth cell type by cNMF, mesenchymal, which most closely resembles tRG in our roadmap. While this resemblance served the purpose of properly separating cancer cell types, mesenchymal cancer cells lack expression of important tRG genes such as AQP4, FAM107A, SOX9, and GLI3, and tRG lack the expression of mesenchymal genes such as CD44 and TIMP1. Therefore, tRG may not be a perfect parallel for mesenchymal cancer cells, and this parallel may not exist in the normal brain. These four cancer cell lineages closely resemble the signatures recently described by Neftel *et al.*⁴⁴, showing their fundamental importance in describing cancer heterogeneity.

However, this is false. FAM107A is expressed by mesenchymal GSCs and in fact mediates mitotic somal translocation and white-matter invasion in both GSCs and in outer radial glia, Bhaduri Cell Stem Cell 2020, Pollen Cell 2016. Likewise, CD44 is expressed by tRG and other radial glia, see above plot from UCSC Cell browser (data from Nowakowski *et al.*). Again, cherry-picking the data and lack of citations to current, relevant literature leaves the impression that the author's conclusion of a GPC as the glioblastoma cell of origin is preconceived.

4. The authors must make a direct bioinformatics comparison to the recent data of Neftel Cell 2019 and Wang Cancer Discovery 2019. The main conclusion of Couturier *et al.* simultaneously contradicts both the findings of Neftel 2019 and Wang 2019. The authors postulate a GSC hierarchy where Neftel 2019 postulates stochastic subtype switching. But, the authors postulate a GSC hierarchy that is the exact opposite of Wang *et al.* Which of Neftel 2019 and Wang 2019 is correct is an open, clinically relevant problem (Fine Cancer Discovery 2019, Platten Neuro-Oncology 2020). I would encourage the authors to place their combined analysis in the context of the Fine and Platten reviews.

5. In their revision the authors say:

hierarchy with progenitor cancer cells at the apex. We found that this progenitor population

contains the majority of the cancer's cycling cells, corresponds to the **apex** of the hierarchy using RNA velocity, and functionally resemble GSCs. Clinically relevant, we show that this hierarchal map can be used to identify therapeutic targets specific to GSCs.

However, this is not supported by their own data. There are many velocity fields in the manuscript which have mesenchymal sources, e.g. below. The authors cherry-pick their data to support a hypothesis of a GPC as the GBM cell of origin.

By the way, this type of crossing of a vector field is impossible. As I have mentioned in my previous review, the RNA-velocity analysis needs a lot of work. I recommend a simplified approach like that taken in Wang 2019.

6. In their rebuttal, the authors say:

We agree with the reviewer that there is a close parallel between proneural GSCs and the progenitors we describe. However, our work, and the work of Neftel et al.¹, identifies multiple cancer cell types which in fact together make this signature, neuronal cancer cells and oligo-lineage cancer cells (both Neftel et al. and our work), as well as progenitors cancer cells (our work). Our sorting paradigm attempts to separate these cell types with the added use of CD24, rather than obtaining the mixed proneural GSC pool using only CD133.

I don't understand what this means. It sounds like the authors are agreeing with me.

The authors compare CD9+/CD44+/CD133- to CD9+/CD133+. But, the CD9+/CD133+ sort will contain CD44+ mesenchymal stem cells. CD133 is the original stem cell marker from King et al. so it is not surprising that CD133+ cells are more stem-like than CD133- cells, but CD133 expression is not exclusive of the mesenchymal phenotype or of CD44 expression (did the authors consider CD44+/CD133+ cells?). As it stands the in vitro and in vivo section is misleading. The text describes the results as concerning the "progenitor" population. However, the results are based on the CD133+ population which is not the same thing. Moreover, the authors have clearly not met the typical standards for GSC characterization.

7. The authors claim that GPCs are at the apex of a GBM cellular hierarchy. However, they also claim that GPCs represent over 25% of the cells in their tumor samples, Figure S4a. GSCs do not make up 25% of a GBM's cells.

REVIEWERS' COMMENTS:

Reviewer #3 (Remarks to the Author):

This is a really interesting study that provides a valuable dataset to the community and offers some real insight, owing to the careful analyses of the authors, into cell types in GBM and how our expanding knowledge in this area may one day alter treatment options, and effectiveness, for patients.

Having undergone several rounds of review at another journal, I find the amends made by the authors for this submission to Nature Communications to be both thorough and informative.

Response to reviewer #2

I commend the authors on their willingness to reevaluate their interpretation. In the revised analysis, the authors have chosen to include truncated radial glia, but not outer radial glia, or other radial glia cell types in their revised map. The authors continue to cherry-pick their analysis to support the hypothesis of an oligodendrocyte progenitor, and not a radial glia, as the glioblastoma cell of origin. This approach ignores much of their own data, as well as several recent important studies. I would ask the authors to consider the following points, which call into question the manuscript's technical correctness, in its current form:

1. The authors must compare the mesenchymal GBM signature to their developing brain data via a simple heatmap on an absolute scale, including the canonical mesenchymal GBM markers: CD44, CHI3L1, NAMPT, TNC, VIM. The “developmental roadmap” projection strategy is needlessly complex and hides the correlation between radial glia and the mesenchymal GBM signatures which a simple heatmap will reveal. I anticipate that the authors will find similar enrichments in their data as are found in the human fetal-brain data of Nowakowski et al. (below), which the authors used for mapping cell labels to their own data.

Comment 1 is an example of trying to force the bias that mesenchymal cancer cells are derived from radial glia. We have shown the relationship between mesenchymal cancer cells and fetal radial glia in the nearest neighbour and roadmap analyses (Figure 3). As we state in the manuscript, important mesenchymal markers like TIMP1, S100A11, and CHI3L1 are not expressed in radial glia. CD44 is one of the most highly expressed genes in mesenchymal cancer cells, yet the expression of CD44 in the Nowakowski dataset, as well as our much larger fetal dataset, is close to background levels. Of note, the recent inclusion of these radial glia in the roadmap analysis, as requested by this reviewer, did not change our main conclusions.

The authors must include all radial glia cell types in their “development roadmap”, including those labeled as RG and uRG by the authors, if they choose to retain their projection strategy at all. By only including tRG, the authors have specifically depleted genes co-expressed by both mesenchymal GBM stem cells and neural stem cells. This is clear from Fig. S4a-b, which shows that tRG are relatively depleted in GSC-enriched cultures while uRG are relatively enriched (yet excluded from the roadmap). Beyond the cherry-picking, the projection approach is problematic overall, it is not clear why the cells were down-sampled, how the cells chosen were selected, and why that number of cells was used. I recommend the authors consider a simpler approach such as a heat map comparison between GBM cell-type signature genes (identified via simple PCA or clustering) and their fetal brain data.

The issues raised in comment 2 were previously raised by this reviewer and thoroughly addressed in previous resubmissions. For example, we explained that the fetal data were down-sampled to have an equal number of all cell types in order to avoid having the fetal neurons over-represented in the roadmap. The allegation of cherry-picking is not correct. We have characterized the cancer signatures in several ways. We used an unbiased method (cNMF) to find cancer signatures (Fig. 1e). We then compared the fetal cells and the cancer cells using a nearest neighbour approach to determine which fetal clusters most resemble the cancer cells (Fig. 3) and confirm this corresponds to these signatures by direct comparison of the fetal and cancer signatures (Supplementary Fig. 4f). We cannot justify including additional radial glial cell types since all cancer signatures are already accounted for by their closest fetal parallel. Finally, the roadmap allows us to visualize the data hierarchically and to clearly attribute cell types. This would not be possible using a heatmap.

However, this is false. FAM107A is expressed by mesenchymal GSCs and in fact mediates mitotic somal translocation and white-matter invasion in both GSCs and in outer radial glia, Bhaduri Cell Stem Cell 2020, Pollen Cell 2016. Likewise, CD44 is expressed by tRG and other radial glia, see above plot from UCSC Cell browser (data from Nowakowski et al.). Again, cherry-picking the data and lack of citations to current, relevant literature leaves the impression that the author's conclusion of a GPC as the glioblastoma cell of origin is preconceived.

Comment 3 regarding FAM107A is similar in nature to comments 1 and 2, forcing the notion that mesenchymal cancer cells are derived from radial glia. My group has been working on FAM107A since 2005. We are the group that showed that it drives glioblastoma cell invasion and provided mechanistic insights (Oncogene 2010, Oncogene 2014). I mention this because we continue to work on this family of genes/proteins and specifically searched for its expression in our dataset. We do not see expression of it in mesenchymal cancer cells. This is strikingly different to its expression in fetal radial glia where it is robustly detected by scRNA.

4. The authors must make a direct bioinformatics comparison to the recent data of Neftel Cell 2019 and Wang Cancer Discovery 2019. The main conclusion of Couturier et al. simultaneously contradicts both the findings of Neftel 2019 and Wang 2019. The authors postulate a GSC hierarchy where Neftel 2019 postulates stochastic subtype switching. But, the authors postulate a GSC hierarchy that is the exact opposite of Wang et al. Which of Neftel 2019 and Wang 2019 is correct is an open, clinically relevant problem (Fine Cancer Discovery 2019, Platten Neuro-Oncology 2020). I would encourage the authors to place their combined analysis in the context of the Fine and Platten reviews.

Comment 4 is seeking comparisons to datasets that have emerged after our original submission to Nature Genetics in March 2019. Reviewer 3 had requested this comparison to Neftel (as well as the TCGA) in the previous submission and these were included in the last resubmission (Supplementary Fig. 4f and g). It showed great similarities between our signatures and those of Neftel et al. We do not believe comparisons to additional studies would further help support the conclusions of our work, but only further delay the publication of this manuscript.

5. However, this is not supported by their own data. There are many velocity fields in the manuscript which have mesenchymal sources, e.g. below. The authors cherry-pick their data to support a hypothesis of a GPC as the GBM cell of origin.

Comment 5 is again related to the reviewer's bias toward the model that mesenchymal cancers cells are the cancer cell of origin. We believe all of the concerns with the velocity analysis, an analysis we performed following the request by this reviewer, were addressed in previous reviews. In the manuscript, we state that in occasional areas, mesenchymal cancer cells appear to be an origin of a vector field; however, the overwhelming trend is for the origin of the vector field to originate within progenitor cancer cell areas.

6. I don't understand what this means. It sounds like the authors are agreeing with me. The authors compare CD9+/CD44+/CD133- to CD9+/CD133+. But, the CD9+/CD133+ sort will contain CD44+ mesenchymal stem cells. CD133 is the original stem cell marker from King et al. so it is not surprising that CD133+ cells are more stem-like than CD133- cells, but CD133 expression is not exclusive of the mesenchymal phenotype or of CD44 expression (did the authors consider CD44+/CD133+ cells?). As it stands the in vitro and in vivo section is misleading. The text describes the results as concerning the "progenitor" population. However, the results are based on the CD133+ population which is not the same thing. Moreover, the authors have clearly not met the typical standards for GSC characterization.

Comment 6. This reviewer has commented on this in each review, and each time we have addressed it. The TCGA signature (including proneural) and the cancer cell types described in our manuscript and by Neftel et al. are related but distinct concepts.

The authors claim that GPCs are at the apex of a GBM cellular hierarchy. However, they also claim that GPCs represent over 25% of the cells in their tumor samples, Figure S4a. GSCs do not make up 25% of a GBM's cells

Comment 7. The proportion of GPC-like cancer cells varies considerably from one patient to another, as shown in Figure 5. While we do not equate GSCs and GPC-like cancer cells, the belief that progenitor cancer cells are always rare within a GBM sample is not supported by our data.

Furthermore, GSCs are determined functionally, since there isn't a transcriptomics cut-off to discern the transitions of cell states – stem versus differentiated. One of the main points of this study is that there is a hierarchy in glioblastoma, and there is also a hierarchy within the stem cell pool, as our in vivo and in vitro data show. We called cells GSCs when they met classical criteria: growth in serum free media, growth as spheres, and tumor forming capacity in PDXs. We cannot quantify the percentage of GSC with a tumor from transcriptomics.